# KNOWLEDGE-DRIVEN SCENE PRIORS FOR SEMANTIC AUDIO-VISUAL EMBODIED NAVIGATION

## ABSTRACT

Generalisation to unseen contexts remains a challenge for embodied navigation agents. In the context of semantic audio-visual navigation (SAVi) tasks, the notion of generalisation should include *both* generalising to unseen indoor visual scenes as well as generalising to unheard sounding objects. However, previous SAVi task definitions do not include evaluation conditions on truly novel sounding objects, resorting instead to evaluating agents on unheard sound clips of known objects; meanwhile, previous SAVi methods do not include explicit mechanisms for incorporating domain knowledge about object and region semantics. These weaknesses limit the development and assessment of models' abilities to generalise their learned experience. In this work, we introduce the use of knowledge-driven scene priors in the semantic audio-visual embodied navigation task: we combine semantic information from our novel knowledge graph that encodes object-region relations, spatial knowledge from dual Graph Encoder Networks, and background knowledge from a series of pre-training tasks—all within a reinforcement learning framework for audio-visual navigation. We also define a new audio-visual navigation sub-task, where agents are evaluated on novel sounding objects, as opposed to unheard clips of known objects. We show improvements over strong baselines in generalisation to unseen regions and novel sounding objects, within the Habitat-Matterport3D simulation environment, under the SoundSpaces task. We will release all code, knowledge graphs, and pre-training datasets upon acceptance.

## 1 INTRODUCTION

Humans are able to use background experience, when navigating unseen or partially-observable environments. Prior experience informs their world model of the semantic relationships between objects commonly found in an indoor scene, the likely object placements, and the properties of the sounds those objects emit throughout their object-object and object-scene interactions. Artificial embodied agents, constructed to perform goal-directed behaviour in indoor scenes, should be endowed with similar capabilities; indeed, as autonomous agents enter our homes, they will need intuitive understanding about how objects are placed in different regions of houses, for better interaction with the environment. Whereas external (domain) knowledge can yield improvements in agent sample-efficiency while learning, generalisability to unseen environments during inference, and overall interpretability in its decision-making, the goal of finding generalisable solutions by injecting knowledge in embodied agents remains elusive (Oltramari et al., 2020; Francis et al., 2022).

The task of semantic audio-visual navigation (shown in Fig. 1) lends itself especially well to the use of domain knowledge, e.g., in the form of human-inspired background experience (encapsulated as a prior over regions and semantically-related objects contained therein). Certain sounds can be associated with particular places, e.g., a smoke alarm is more likely to originate in the kitchen. To infer such semantic information from sounds in an environment, we propose the idea of a knowledge-enhanced prior.

By using a prior enriched with general experiences, we hypothesise that the learned model would generalise to novel sound sources. We adopt a modular training paradigm, which has been shown to lead to improvements in cross-domain generalisability and more tractable optimisation (Chen et al., 2021b; Chaplot et al., 2020b; Francis et al., 2022). To verify our hypotheses, we evaluate the agent's performance on a set of novel sounding objects that were not introduced during training.

**Contributions.** First, we introduce the use of knowledge-driven scene priors in the semantic audio-visual embodied navigation task: we combine semantic information from our novel knowledge graph that encodes object-region relations, spatial knowledge from dual Graph Encoder Networks, and background knowledge from a series of pre-training tasks—all within a reinforcement learning (RL) framework. Second, we define a knowledge graph that encodes object-object, object-region, and region-region relations in house environments. Next, we curate a multimodal dataset for pre-training a visual encoder, in order to encourage object-awareness in visual scene understanding. Finally, we define a new task of semantic audio-visual navigation, wherein we assess agent performance on the basis of their generalisation to truly novel sounding objects. We offer experimental results against strong baselines, and show improvements over these models on various performance metrics in unseen contexts. We will provide all code, dataset-generation utilities, and knowledge graphs upon acceptance of the manuscript.

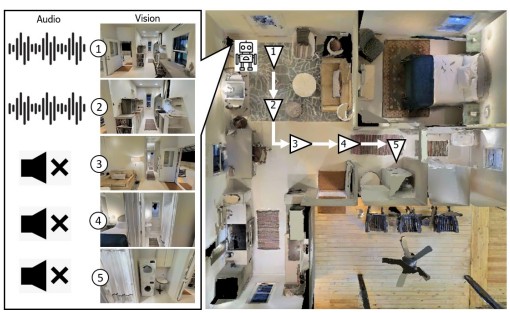

Figure 1: Illustration of the proposed semantic audio-visual navigation task. The agent is initialised at a random location in a 3D environment and tasked to navigate to the sounding object based on audio and visual signals. The sound signal may stop while the agent is navigating (e.g., sound produced by washing machine stops). Thus, the agent is encouraged to understand the sound and visual semantics to reason about where to search for the sounding object. For example, in the image above, the agent hears the washing machine sound and decides to navigate near the bathroom to search for the washing machine.

## 2 RELATED WORK

**Modularity in goal-driven robot navigation.** Goal-oriented navigation tasks have long been a topic of research in robotics (Kavraki et al., 1996; Lavalle et al., 2000; Canny, 1988; Koenig & Likhachev, 2006). Classical approaches generally tackle such tasks through non-learning techniques for searching and planning, e.g., heuristic-based search (Koenig & Likhachev, 2006) and probabilistic planning (Kavraki et al., 1996). Although classical approaches might offer better generalisation and optimality guarantees in low-dimensional settings, they often assume accurate state estimation and cannot operate on high dimensional raw sensor inputs (Gordon et al., 2019). More recently, researchers have pursued data-driven techniques, e.g., deep reinforcement learning (Wijmans et al., 2020; Batra et al., 2020; Chaplot et al., 2020a; Yang et al., 2019; Chen et al., 2021b;a; Gan et al., 2020) and imitation learning (Irshad et al., 2021; Krantz et al., 2020), to design goal-driven navigation policies. End-to-end mechanisms have proven to be powerful tools for extracting meaningful features from raw sensor data, and thus, are often favoured for the setting where agents are tasked with learning to navigate toward goals in unknown environments using mainly raw sensory inputs. However, as task complexity increases, these types of systems generally exhibit significant performance drops, especially in unseen scenarios and in long-horizon tasks (Gordon et al., 2019; Saha et al., 2021). To address the aforementioned limitations, modular decomposition has been explored in recent embodied tasks. Chaplot et al. (2020c) design a modular approach for visual navigation, consisting of a mapping module, a global policy, and a local policy, which, respectively, builds and updates a map of the environment, predict the next sub-goal using the map, and predicts low-level actions to reach the sub-goal. Irshad et al. (2021) also define a hierarchical setup for Vision-Language Navigation (VLN) (Anderson et al., 2018), where a global policy performs waypoint-prediction, given the observations, and a local policy performs low-level motion control. Gordon et al. (2019) design a hierarchical controller that invokes different low-level controllers in charge of different tasks such as planning, exploration, and perception. Similarly, Saha et al. (2021) design a modular mechanism for mobile manipulation that decomposes the task into: mapping, language-understanding, modality grounding, and planning. Aforementioned modular designs have shown to increase task performance and generalisability, especially in unexplored scenarios, compared to their end-to-end counterparts. Motivated by these, we develop a modular framework for semantic audio-visual navigation, which includes pre-trained and knowledge-enhanced scene priors, enabling improved unseen generalisation.

**Knowledge graphs in visual navigation.** Combining prior knowledge with machine learning systems remains a widely-investigated topic in various research fields, such as natural language processing (Ma et al., 2021; 2019; Francis et al., 2022), due to the improvements in generalisability and sample-efficiency that symbolic representation promises for learning-based approaches.

Historically, integrating symbolic knowledge with, e.g., navigation agents has proven non-trivial, yielding a collection of research areas focusing on smaller components of the problem—such as finding the appropriate representation of the knowledge (e.g., logical formalism, knowledge graphs, probabilistic graphical models), the appropriate *type* of knowledge that should be encoded (e.g., spatial commonsense, declarative facts, etc.), and the best knowledge-injection mechanism (e.g., graph convolutional networks, grounded natural language, etc.) (Ma et al., 2019). Knowledge graphs have gained popularity due to their interpretability and general availability as existing large-scale resources, such as ConceptNet (Speer et al., 2017) and VisualGenome (Krishna et al., 2016). Fortuitously, graph processing of structured data has experienced a surge of popularity in deep learning in recent years, leading to renewed interest in this neuro-symbolism (Oltramari et al., 2020; Wu et al., 2021). Some visual navigation works exploit knowledge graphs in the pursuit of generalisation (Moghaddam et al., 2020; Yang et al., 2019; Lv et al., 2020; Du et al., 2020; Vijay et al., 2019). Yang et al. (2019) create knowledge graphs based on VisualGenome (Krishna et al., 2016) and inject features extracted from the graph as prior knowledge in visual navigation. In similar fashion, Qiu et al. (2020) provide agents with knowledge of object relational semantics. However, the priors provided by these works only leverage object-object connections. Lv et al. (2020) show improvements in goal-directed visual navigation by injecting 3D spatial knowledge into learning-based agents. Inspired by these works, we construct a knowledge graph that includes *all* object-object, object-region, and region-region declarative semantics, which enables the more complex reasoning path, *sound → object → region*, in audio-visual navigation. Therefore, to our best knowledge, we become the first to study knowledge-driven scene priors for the audio-visual navigation task family.

**Generalization to unseen contexts.** Chen et al. (2020; 2021b;a) leverage the SoundSpaces (Chen et al., 2020) simulation environment and dataset to design and assess Audio-Visual Navigation policies. The dataset is based on photorealistic indoor environments from the Matterport3D (Chang et al., 2017) and Replica (Straub et al., 2019) datasets, to which 102 sound sources commonly found in indoor environments (e.g., household appliances, musical instruments, telephones, etc.) were incorporated. The SoundSpaces dataset is split, such that indoor scenes encountered during testing are not found in the episodes used during the training stage. However, sounds of objects encountered during training may also appear during testing. Gan et al. (2020) also explore Audio-Visual Navigation, but using the simulation platform AI2-THOR (Kolve et al., 2017), which contains computer-generated graphical imagery. The authors introduce the Visual-Audio Room (VAR) benchmark consisting of seven different indoor environments—two of which were used for training and five for testing. The VAR benchmark incorporates three different audio categories: ring tone, alert alarm, and clocks. Similar to the AVN task introduced before, the same sound sources are found both in the training scenes, as well as and the testing scenes. In this paper, we argue that in the context of Audio-Visual Navigation tasks, generalisation to unseen environments pertains to both generalising to unseen visual scenes, as well as to unheard sounds. Current Audio-Visual benchmarks do not take into consideration the latter. Thus, there is no direct assessment of generalisation performance to unheard sounds. To tackle this limitation, we propose a curated version of the SoundSpaces dataset where we evaluate our agent in four conditions: (1) seen houses and heard sounds, (2) seen houses and unheard sounds, (3) unseen houses and heard sounds, and (4) unseen houses and unheard sounds.

## 3 PROBLEM DEFINITION

We first consider the semantic audio-visual navigation (SAVi) task (Chen et al., 2021a): an agent is initialised at a random location of an unmapped 3D house environment, which contains a sounding object (e.g., piano). The agent must reach the sounding object, using its sensory inputs, consisting of vision and audio. Two assumptions are made in this task: firstly, the target sound has variable length in an episode and may not be available at every time step; the sound (e.g., a telephone ringing) may stop during navigation; secondly, the sounding object has a physical and semantically-meaningful embodiment in the scene (e.g., the sound of a telephone ring is associated with a physical manifestation of a telephone, as opposed to the sound of an airplane passing overhead being associated with the center of the living room). These assumptions are realistic because sound events have variable length in the real world and are based on the semantics of the corresponding sounding objects. Due to the variable-length nature of the sound, the agent cannot rely *exclusively* on the audio signal to reach the sounding object: instead, the agent must use the audio signal to both predict the sounding object's location as well as understand the object's semantics. Moreover, the agent needs to associate its visual cues with the sound and reason about object and region relationships, in order to navigate effectively.

To study these phenomena, we extend the SAVi task by evaluating agents on completely unheard sounding objects. In the original task (Chen et al., 2021a), agents were evaluated on unheard clips of *known* sounding objects, whereas in our task, agents are evaluated on completely unknown sounding objects. More formally, we consider a set of sounding objects $\mathcal{O}$ (e.g., shower, TV monitor, etc.), a set of indoor regions $\mathcal{R}$ (e.g., bathroom, living room), and a set of houses $\mathcal{H}$. A particular house $h_i \in \mathcal{H}$ has a set of regions $\{r_{i1}, r_{i2}, \ldots, r_{ij}\}$ and a set of objects $\{o_{i1}, o_{i2}, \ldots, o_{ik}\}$, where there are $k$ objects placed in $j$ regions of the house $h_i$. Note that there are multiple instances of each sounding object $o \in \mathcal{O}$ and region $r \in \mathcal{R}$ across all houses $\mathcal{H}$. We divide the total set of possible houses $\mathcal{H}$ into two mutually exclusive subsets: $\mathcal{H}_{seen}$ and $\mathcal{H}_{unseen}$. Similarly, we divide sounding objects $\mathcal{O}$ into $\mathcal{O}_{heard}$ and $\mathcal{O}_{unheard}$. The houses in $\mathcal{H}_{seen}$ and the sounding objects in $\mathcal{O}_{heard}$ are only experienced by agents during training; agents are evaluated on unheard sounding objects $\mathcal{O}_{unheard}$. To solve this task, agents must learn to reason about the novel sounds based on prior knowledge; our work aims to enable agents to reach sounding objects they have never experienced before.

## 4    KNOWLEDGE-DRIVEN SCENE PRIORS FOR AUDIO-VISUAL NAVIGATION

We introduce a **k**nowledge-driven approach for **s**emantic **a**udio-**v**isual **e**mbodied **n**avigation (`K-SAVEN`), which incorporates scene priors in knowledge graph form and extracts relational features using Graph Encoder Networks (GEN) (Kipf & Welling, 2017) for audio and visual modalities. GENs provide agents with reasoning capability, using prior knowledge, and dynamically update their beliefs according to new observations. Our model also incorporates Scene Memory Transformer (SMT) (Fang et al., 2019) that captures long-term dependencies by recording visual features in memory and locating the goal by attending to acoustic features. We compute visual features by combining a vision-based semantic knowledge vector with visual encoder representations. Similarly, we use audio observations to compute acoustic features, combining audio-based semantic knowledge vector, features encoded from the audio encoder, and location prediction. Thus, the prior knowledge-driven reasoning capability using GENs with the memory-based attention mechanism using SMT allows the agent to generalise to novel houses and sounding objects, exploit spatio-temporal dependencies, and efficiently navigate to goal. The 6 modules of `K-SAVEN` are summarised in Fig. 2: 1) Pre-trained models that, given the audio and visual observations from the environment, predict objects and regions; 2) Graph Encoder Networks that compute audio-semantic and visual-semantic feature embeddings; 3) Vision Encoder that projects the visual observations at each step to an embedding space; 4) Audio Encoder that projects the audio observations at each step to an embedding space; 5) Location Predictor that, given the acoustic signal from the sounding object, predicts its relative distance and direction from the agent; 6) Scene Memory Transformer that uses an attention-based policy network, which computes a distribution over actions, given the encoded observations in scene memory and the acoustic observation that captures goal information. We detail each module below.

**Modular Pre-training.** In our task, the agent relies on audio observations to set its goal and uses visual observations to navigate to that goal; the agent must detect objects and regions in a given observation. To this end, we trained audio classification model $f_c^b$ to predict a score for each object $o \in \mathcal{O}$, as likelihood that $o$ produced the acoustic observation, and a vision classification model $f_c^v$ to predict a score for each object $o \in \mathcal{O}$ and region $r \in \mathcal{R}$ as likelihood that the observation corresponds to region $r$. The acoustic event has variable length and may not be present at each time step, so the agent cannot rely on the current audio observation alone as a persistent signal. Thus, our model aggregates the current prediction $\hat{c}_t^b$ with the previous prediction $c_{t-1}^b$, $c_t^b = f_\delta(\hat{c}_t^b, c_{t-1}^b) = (1 - \delta)\hat{c}_t^b + \delta c_{t-1}^b$, where $\delta$ is the weighting factor set to 0.5. When the acoustic event stops (i.e., zero sound intensity), the agent uses its latest estimate $c_t^b$.

**Knowledge graph construction.** Our knowledge graph captures spatial relationships between object-to-object, object-to-region, and region-to-region. This prior knowledge about how objects are placed in regions of houses enables the agent to reason about where to find novel-sounding objects for efficient navigation; more precisely, this prior knowledge enables the reasoning path, *Sound →
Object → Region*, which is crucial to the task of audio-conditioned visual navigation. For example, suppose the squeaky sound produced by a chair is novel to the agent, and it knows that chairs are usually kept close to tables or cushions and found in living rooms, or offices. In that case, it may decide to navigate to regions that usually have chairs and objects usually placed close to chairs, which would lead to finding the chair faster than not knowing such spatial and semantic relationships between objects and regions. Our knowledge graph is denoted by an undirected graph $G = (V, E)$, where $V$ and $E$ denote vertices and edges, respectively. Each vertex denotes an object or region, and

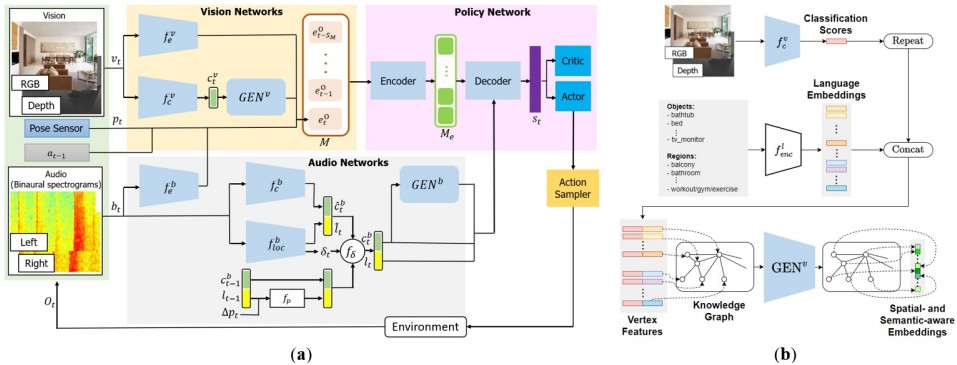

Figure 2: **(a) K-SAVEN's system overview.** Visual observation $v_t$ is fed to two modules: vision encoder $f_e^v$, which encodes the visual observation, and pre-trained vision model $f_c^v$, which, given the visual observation, predicts classification scores $c_t^v$ for objects and regions. These scores are used by the vision-based graph encoder network $GEN^v$ to compute visual-semantic feature embeddings. Binaural audio observation $b_t$ is fed to three models: audio encoder $f_e^b$ (encodes the audio observation), location predictor $f_{loc}^b$ (predicts distance and direction $l_t$ of the sounding object from the agent, and direct-to-reverberant-ratio $\delta_t$), and pre-trained audio model $f_c^b$ (given the audio observation, predicts classification scores $c_t^b$ for objects). These scores are used by the audio-based graph encoder network $GEN^b$ to compute audio-semantic feature embeddings. The outputs of $f_e^v$, $GEN^v$ and $f_e^b$ are stored in memory $M$, along with the agent's pose $p_t$ and previous action $a_{t-1}$. The attention-based policy network conditions the encoded visual information $M_e$ on the acoustic information, enabling the agent to associate visual cues with acoustic events and predict the state representation $s_t$, which contains spatial and semantic cues helpful to reach the goal faster. The actor-critic network, given the state $s_t$, predicts the next action $a_t$. **(b) Graph Encoder Networks.** Each vertex denotes an object or region category. The initial vertex features fed into the $GEN^v$ are initialised with the joint embedding obtained by concatenating word embeddings of object or region names and classification scores of objects and regions based on the current observation. $GEN^v$ performs information propagation through the three layers, and the output of the $GEN^v$ is spatial and semantic aware embeddings. The audio-based GEN uses $f_c^b$ and $GEN^b$ instead of $f_c^v$ and $GEN^v$.

each edge denotes the relationship between a pair of vertices. To compute these relationships, we use Matterport3D dataset (MP3D; Chang et al. (2017)) as it contains semantic labels of 42 objects and 30 regions for 90 houses. We only use 21 objects and 24 regions ($|V| = 45$), which were used in the original SAVi task (Chen et al., 2021a) to build the knowledge graph (more details in Section 5). More specifically, two objects are connected with an edge if they are found in the same region, and their frequency of occurrence is above a threshold. We compute this frequency with respect to the most frequent object of that region and set the threshold to the maximum value that connects each object with at least one other object. An object and region are connected if the region contains other objects, which are connected with the object based on object-to-object relations. Finally, two regions are connected if their frequency of containing connected objects, based on object-to-object relations, is above a threshold. We set the threshold to the maximum value connecting each region with at least one other region. Further knowledge graph construction and representation details are in Appendix B.

**Location Prediction and Direct-to-Reverberant Ratio Estimation.** The audio observation contains information about the relative distance and direction from the agent to the sounding object. Thus, we jointly trained a location predictor $f_{loc}^b$ to predict a location $\hat{l}_t = (\Delta x, \Delta y)$, relative to the current pose $p_t$ of the agent, and the direct-to-reverberant ratio (DRR) $\in [0, 1]$ of the impulse response between the sounding source and the agent. Similar to the pre-trained audio model, our location prediction also aggregates the current estimate $\hat{l}_t$ with the previous $l_{t-1}$, $l_t = f_\delta(\hat{l}_t, l_{t-1}, \Delta p_t, \delta_t) = \delta_t \hat{l}_t + (1 - \delta_t) f_p(l_{t-1}, \Delta p_t)$, where $f_p(\cdot)$ transforms the previous location prediction $l_{t-1}$ based on the last pose change $\Delta p_t$. Here, $\delta_t$ is either fixed to $0.5 \ \forall t$ (exponential average) or assigned the value of the estimated DRR (dynamic average). The agent uses its latest estimate $l_t = f_p(l_{t-1}, \Delta p_t)$ when the acoustic event stops. Note that DRR prediction also serves as an auxiliary task, as it will help the agent better estimate the directness and location of the sounding object. In fact, DRR provides an indirect measure of the acoustic distance between the source and the agent, independent of the sound level of the source. At training time, we build the ground truth for $\delta_t$ from the room impulse response (RIR) between the source and the agent as the ratio between the energy of the RIR in the first 10ms after the peak and the overall energy of the RIR. Thus, $\delta_t$ measures how direct the acoustic propagation between the sounding object and the agent is: when the agent is far from the source, $\delta_t$

tends towards 0; as the agent gets closer to the source, $\delta_t$ increases. When the source is silent, $\delta_t$ equals 0; thus, $\delta_t$ predicts trustworthiness of location prediction, based on the binaural sound itself.

**Encoder Networks.** The goal of $GEN^v$ and $GEN^b$ are to extract a semantic knowledge vector using the graph $G = (V, E)$. As shown in Fig. 2, the input to each vertex $v$ is feature vector $x_v$, which is a concatenated representation of both semantic cues (i.e., language embeddings) and the visual or acoustic cues (i.e., the classification score for objects and regions based on the current visual image or sound signal). The language embeddings are generated by GloVe (Pennington et al., 2014) ($f_{enc}^l$) and the classification score is generated by pre-trained vision ($f_c^v$) or audio ($f_c^b$) models (see Section 4). The knowledge graph is represented as a binary adjacency matrix $A$. Similar to Yang et al. (2019); Kipf & Welling (2017), we perform normalisation on $A$ to obtain $\tilde{A}$. Let $X = [x_1, \ldots, x_{|V|}] \in R^{|V| \times D}$ be the inputs of all vertices and $Z = [z_1, \ldots, z_{|V|}] \in R^{|V| \times F}$ be the output of the GENs, where $D$ and $F$ denote the dimension of the input and output feature. Our GENs perform the following layer-wise information propagation rule: $H^{(l+1)} = \sigma(\tilde{A}H^{(l)}W^{(l)})$. Here, $H^{(0)} = X, H^{(L)} = Z, W^{(l)}$ is the parameter for the $l$-th layer, $L$ is the number of GEN layers, and $\sigma$ denotes an activation function. We initialise each vertex based on current observation then perform information-propagation to compute audio-based and vision-based semantic knowledge vectors. The vision-based knowledge vector is stored in memory $M$, and the audio-based knowledge vector is used to attend to the encoded memory $M_e$. The output is a graph embedding which serves as a spatial- and semantic-aware representation for policy optimisation. Our vision encoder $f_e^v$ encodes the visual observations, consisting RGB and depth images from the agent's perspective. Our audio encoder $f_e^b$ encodes the binaural audio observations heard by the agent into a two-channel log-mel spectrogram, with a third channel encoding the generalised cross-correlation with phase transform (Knapp & Carter, 1976) between the two channels.

**Policy Network.** We use a transformer-based architecture for our RL policy network, which stores observations in memory $M$. At each time step, our model encodes each visual observation, $e_t^v = f_e^v(v_t)$ and $e_t^{v-gen} = GEN^v(f_c^v(v_t))$ to save in the memory. Our model also stores in memory the agent's pose $p$, defined by its location and orientation $(x, y, \theta)$ with respect to its starting pose $p_0$ in the current episode, and $a_{t-1}$, the previously executed action. Thus, the encoded observation stored in memory is $e_t^O = [e_t^v, e_t^{v-gen}, p_t, a_{t-1}]$. The model stores these observation encodings up to time $t$ in memory: $M = \{e_t^O : i = max\{0, t - S_M\}, \ldots, t\}$, where $S_M$ is the memory size. The transformer uses the memory $M$ stored so far in the episode and encodes these visual observation embeddings with a self-attention mechanism to compute the encoded memory $M_e = Encoder(M)$. Then, using the audio observation embeddings, a decoder network attends to all cells in $M_e$ to calculate the state representation $s_t = Decoder(M_e, e_t^{b-gen}, l_t^b)$, where $e_t^{b-gen} = GEN^b(f_c^b(b_t))$. Using this attention mechanism, the agent captures long-term spatio-temporal associations between the acoustic-driven goal prediction and the visual observations. Moreover, our model preserves the most relevant information to reach the goal by conditioning visual-semantic embeddings stored in $M_e$ on audio-semantic embeddings computed using current audio observation. The actor-critic network uses $s_t$ to predict the value of the state and action distribution. Finally, the action sampler takes next-action $a_t$ from this action distribution.

**Learning and Optimisation.** To train the vision classification model $f_c^v$, we collect a dataset using 85 MP3D houses, consisting of 82,828 images, each corresponding to a location and rotation angle in the SoundSpaces simulator (see Section 5). Each image has 128 x 128 resolution and 4 modalities: RGB image, depth image, object semantic image, and region semantic image. We use the binary cross-entropy loss for optimising the vision classification model and train it as a standard multi-label classifier. To train the audio classification model $f_c^b$, we use the SoundSpaces simulator to generate 1.5M spectrograms using different source and receiver positions, each corresponding to a sounding object in one of the 85 MP3D houses. We treat detecting sounding objects as a multi-class classification problem and optimise the audio classification model using cross-entropy loss. Our vision classification model takes an RGB image as input, and the audio classification model takes 1 second sound clip represented as two $65 \times 26$ binaural spectrograms as input. We trained both vision and audio classification models using a ResNet-18 (He et al., 2015) architecture, pre-trained on ImageNet. The vision classification model predicts a score for 21 objects and 24 regions, and the audio classification model predicts a score for 21 objects (see Section 5). These models are pre-trained before and are frozen during policy optimisation. While we use MP3D, in this paper, for training these classification models, we assert that our modules may also be trained on other house

Table 1: Results of baseline models and our proposed approach.

| Method | SEEN HOUSES, HEARD SOUNDS | | | | | SEEN HOUSES, UNHEARD SOUNDS | | | | |
|---|---|---|---|---|---|---|---|---|---|---|
| | SR (↑) | SPL (↑) | SNA (↑) | DTG (↓) | SWS (↑) | SR (↑) | SPL (↑) | SNA (↑) | DTG (↓) | SWS (↑) |
| Random | 4.7 | 1.0 | 0.4 | 18.3 | 4.7 | 6.8 | 1.9 | 0.9 | 16.3 | 6.7 |
| AudioGoal Chen et al. (2020) | 31.2 | 29.5 | 21.3 | 7.9 | 9.6 | 17.4 | 16.6 | 11.9 | 10.7 | 5.8 |
| AudioObjectGoal | 40.8 | 39.2 | 29.5 | 5.7 | 13.6 | 17.7 | 16.4 | 11.7 | 9.7 | 6.6 |
| SAVi Chen et al. (2021a) | 67.2 | **53.6** | 52.8 | **1.6** | **37.8** | 21.7 | 15.7 | 13.6 | 6.5 | 12.1 |
| K-SAVEN (*ours*) | **70.2** | 52.8 | **53.9** | 1.78 | 31.0 | **37.8** | **27.1** | **25.5** | **5.3** | **17.8** |

| Method | UNSEEN HOUSES, HEARD SOUNDS | | | | | UNSEEN HOUSES, UNHEARD SOUNDS | | | | |
|---|---|---|---|---|---|---|---|---|---|---|
| | SR (↑) | SPL (↑) | SNA (↑) | DTG (↓) | SWS (↑) | SR (↑) | SPL (↑) | SNA (↑) | DTG (↓) | SWS (↑) |
| Random | 6.2 | 1.5 | 0.7 | 17.7 | 6.1 | 5.6 | 1.7 | 0.7 | 14.8 | 5.8 |
| AudioGoal Chen et al. (2020) | 15.7 | 14.9 | 10.7 | 14.6 | 4.2 | 16.5 | 15.5 | 10.4 | 12.8 | 5.6 |
| AudioObjectGoal | 14.9 | 13.9 | 10.2 | 14.2 | 4.6 | 14.3 | 12.9 | 8.7 | 12.2 | 5.5 |
| SAVi Chen et al. (2021a) | 32.0 | 21.2 | 18.5 | 10.1 | 17.9 | 15.3 | 10.8 | 8.8 | 10.0 | 8.3 |
| K-SAVEN (*ours*) | **35.3** | **24.4** | **22.2** | **8.4** | **18.6** | **34.4** | **23.4** | **21.7** | **6.6** | **14.3** |

Table 2: Results of various ablative model configurations, across all dataset splits.

| Method | SEEN HOUSES, HEARD SOUNDS | | | | | SEEN HOUSES, UNHEARD SOUNDS | | | | |
|---|---|---|---|---|---|---|---|---|---|---|
| | SR (↑) | SPL (↑) | SNA (↑) | DTG (↓) | SWS (↑) | SR (↑) | SPL (↑) | SNA (↑) | DTG (↓) | SWS (↑) |
| SAVi Chen et al. (2021a) | 67.2 | 53.6 | 52.8 | 1.6 | 37.8 | 21.7 | 15.7 | 13.6 | 6.5 | 12.2 |
| K-SAVEN *−only $GEN^b$* | 64.4 | 52.5 | 50.1 | 2.1 | 38.0 | 31.7 | 23.2 | 22.2 | 5.7 | 15.6 |
| K-SAVEN *−only $GEN^v$* | 73.2 | 58.7 | 61.1 | 1.6 | 39.4 | 29.7 | 21.8 | 20.7 | 6.2 | 14.9 |
| K-SAVEN *−both $GENs$* | 73.0 | 58.6 | 58.8 | **1.3** | 39.4 | 30.5 | 22.6 | 21.8 | 6.0 | 16.0 |
| K-SAVEN *−both $GENs + \delta_t$* | 66.6 | 49.5 | 48.2 | 1.8 | 36.2 | 34.7 | 24.8 | 24.8 | 5.8 | 14.0 |
| K-SAVEN *−full model* | 70.2 | 52.8 | 53.9 | 1.78 | 31.0 | 37.8 | 27.1 | 25.5 | 5.3 | 17.8 |

| Method | UNSEEN HOUSES, HEARD SOUNDS | | | | | UNSEEN HOUSES, UNHEARD SOUNDS | | | | |
|---|---|---|---|---|---|---|---|---|---|---|
| | SR (↑) | SPL (↑) | SNA (↑) | DTG (↓) | SWS (↑) | SR (↑) | SPL (↑) | SNA (↑) | DTG (↓) | SWS (↑) |
| SAVi Chen et al. (2021a) | 32.0 | 21.2 | 18.5 | 10.1 | 18.0 | 15.3 | 10.8 | 8.8 | 10.0 | 8.3 |
| K-SAVEN *−only $GEN^b$* | 31.1 | 21.3 | 19.6 | 9.8 | 15.1 | 23.3 | 16.1 | 14.8 | 9.5 | 10.0 |
| K-SAVEN *−only $GEN^v$* | 32.8 | 23.2 | 21.1 | 9.4 | 16.0 | 21.2 | 14.2 | 12.4 | 9.3 | 10.0 |
| K-SAVEN *−both $GENs$* | 31.9 | 21.7 | 20.1 | 10.0 | 16.0 | 22.9 | 15.3 | 13.7 | 9.2 | 10.1 |
| K-SAVEN *−both $GENs + \delta_t$* | 29.8 | 19.9 | 17.9 | 9.5 | 13.9 | 27.2 | 17.2 | 16.5 | 8.5 | 10.2 |
| K-SAVEN *−full model* | **35.3** | **24.4** | **22.2** | **8.4** | **18.6** | **34.4** | **23.4** | **21.7** | **6.6** | **14.3** |

environments that provide semantic labels of objects and regions in houses. For location predictor $f_{loc}^b$, we use a simplified ResNet-18 architecture and train it jointly with the policy, using the same experience. We optimise the location predictor using the mean-squared error loss and update it with the same frequency as the policy network. We train the policy network using the decentralised distributed proximal policy optimisation (DD-PPO) (Wijmans et al., 2020), which consists of a value network loss, policy network loss, and an entropy loss to encourage exploration (Schulman et al., 2017). We adapt the two-stage training procedure proposed by Fang et al. (2019) for effectively training the vision networks ($f_e^v$, $GEN^v$). In the first stage, the SMT policy is trained without attention by setting the memory size $s_M = 1$ and storing the latest observation embeddings. In the second stage, the memory size is set to $s_M = 150$, and the parameters of the vision networks are frozen. The input to the vision encoder $f_e^v$ is $64 \times 64$ RGB, and depth images cropped from the center. We optimise our model using Adam (Kingma & Ba, 2015) with a learning rate of $2.5 \times 10^{-4}$ for the policy network and $1 \times 10^{-3}$ for the pre-trained audio and vision networks using PyTorch (Paszke et al., 2019). We train our method and the baselines for 300M steps and roll out policies for 150 steps. See Appendix C for more details.

## 5 EXPERIMENTS

**Simulator and semantic sounds.** We use SoundSpaces (Chen et al., 2020) to simulate an agent navigating in visually- and acoustically-realistic 3D house environments. While, SoundSpaces supports two real-world environment scans (Replica (Straub et al., 2019) and Matterport3D (MP3D) (Chang et al., 2017)), we used MP3D as it provides a larger number of houses and object-region semantics therein. We use the same 21 object categories as Chen et al. (2021a) for MP3D; these object categories are visually present in the 24 regions of the 85 MP3D houses. We use the publicly-available sound clips from the experiment performed by Chen et al. (2021a), in which audio clips from `freesound.org` database were used. We generate sound by rendering the specific sound that semantically matches the object at the locations in MP3D houses. For example, the water-dropping sound will be associated with the sink in the kitchen. See Appendix A and F for more information about object/region categories and episode specification.

**Rewards.** The agent receives a sparse reward of $+10$ when it reaches the goal, a dense reward of $+1$ for reducing the geodesic distance to goal, and an equivalent negative reward for increasing it. To encourage trajectory efficiency, we also assign a reward of $-0.01$ per time step. To avoid simpler episodes, wherein is easy to reach goal (e.g., straight paths or short distance), we used 2 conditions while sampling episodes: 1) the ratio of geodesic distance to euclidean distance must be greater than 1.1; 2) the geodesic distance from the start location to the goal location must be greater than 4 meters.

**Baseline models.** We compare our model against several baselines: *Random walk* is a baseline which uniformly samples one of the three navigation actions with probability of 0.33, or *Stop* with probability 0.01. *Stop* is also executed automatically by the simulator when the agent's location is within 1m radius of the target sounding object, or if more than 500 steps are taken by the agent. *AudioGoal* (Chen et al., 2020) is an end-to-end RL policy based on the PointGoal task (Wijmans et al., 2020) based on a Seq2Seq mechanism which uses a GRU state encoder that leverages colour and depth images to navigate the unknown environments. In contrast to PointGoal, which uses GPS sensing to guide the agent toward its goal, this baseline uses audio spectrograms. *AudioObjectGoal* is a Seq2Seq mechanism similar to *AudioGoal*, but the agent is also provided with the semantic label of the target object. *SAVi* (Chen et al., 2021a) is a transformer-based model that uses a goal descriptor network to predict both spatial and semantic properties of the target sounding object. It is the state-of-the-art deep RL model for the semantic audio-visual navigation task; like `K-SAVEN`, it uses SMT and a pre-trained audio classification model.

**Evaluation metrics.** We follow Chen et al. (2021b;a) in reporting agent performance against the following metrics: 1) success rate (SR); 2) success rate weighted by path length (SPL); 3) success rate weighted by number of actions (SNA); 4) average distance to goal (DTG) on episode success/termination; and 5) success when silent (SWS). We assess model generalisation by evaluating our method on *unheard* sounding objects, across the following settings: 1) seen houses and heard sounds; 2) seen houses and unheard sounds; 3) unseen houses and heard sounds; and 4) unseen houses and unheard sounds. We randomly split the houses and sounding objects for training and testing. We use 68 seen houses, 17 unseen houses, 16 heard sounding objects, and 5 unheard sounding objects; we average the results over 1,000 episodes for each setting.

## 6 RESULTS

**Quantitative results discussion.** The performance comparison between the aforementioned baseline agents—across Seen-House/Heard-Sounds (SH/HS), Seen-House/Unheard-Sounds (SH/US), Unseen-House/Heard-Sounds (UH/HS), and Unseen-House/Unheard-Sounds (UH/US) conditions—is summarised in Table 1. Overall, in all cases except Seen-Houses/Heard-Sounds, our approach outperforms all baseline methods across all metrics. More specifically, in the Seen-Houses/Unheard-Sounds case, there is an improvement of 20.4%, 20.1%, and 16.1% in SR, in the Unseen-Houses/Heard-Sounds, there is an improvement of 19.6%, 20.4%, and 3.3% in SR, and in the Unseen-Houses/Unheard-Sounds case, there is an improvement of 17.9%, 20.1%, and 19.1% in SR as compared to AudioGoal, AudioObjectGoal, and SAVi, respectively. These results indicate that our agent could leverage the reasoning capability using GENs with the memory-based attention mechanism using SMT and generalise to the novel sounding objects. In the Seen-Houses/Heard-Sounds case, where the agent has experienced the sounding objects during training, and it is more critical to reason about the visual cues than the sound semantics to succeed, our approach performs comparable to SAVi. We emphasize that SAVi also has a vision encoder and a scene memory to store encoded vision observations like our approach resulting in comparable performance in the Seen-Houses/Heard-Sounds case with our approach and making it challenging to improve on SAVi with significant margins. Additionally, due to fair comparison, we strictly trained ours and SAVi's models for 300M steps, for both stages.

**Ablations.** We provide ablation results in Table 2, to evaluate our system's key components. Overall, all ablative configurations of our approach perform better than SAVi in all metrics. We note that SAVi also leverages the audio classification model and SMT policy with scene memory, similar to our approach, and as shown in Table 2, adding GEN and direct-to-reverberant (DRR) modules helps to improve the agent's performance further. Our *full model* performs best in all metrics, except for the Seen-Houses/Heard-Sounds (SH/HS) case. These results indicate that our agent can indeed associate visual cues with sound semantics and use the prior knowledge-driven reasoning capability from both GENs to generalise to novel sounds and novel environments to navigate efficiently. Moreover, in the Seen-Houses/Heard-Sounds case, *only-$GEN^v$* outperforms other models in most metrics indicating that $GEN^v$ has a comparatively more significant impact on our model's performance. However, relying exclusively on *only-$GEN^v$* would not enable the agent to navigate to the novel sounds effectively. We evaluate the impact of using the estimated DRR as a weight for location belief update, by comparing `K-SAVEN` –*full model* to `K-SAVEN` –*both GENs* + $\delta_t$, the former using $\delta_t = 0.5$, $\forall t$ (exponential average) and the latter using the estimated DRR as $\delta_t$ (dynamic average). The *full model* achieves better performance by a margin compared to the use of a dynamically-estimated weighting factor $\delta_t$. Our intuition is that DRR-estimation as an auxiliary task for the location-predictor induces better $\hat{l}_t$ estimation, as DRR acts as a proxy to the estimation of the distance to the source. However,

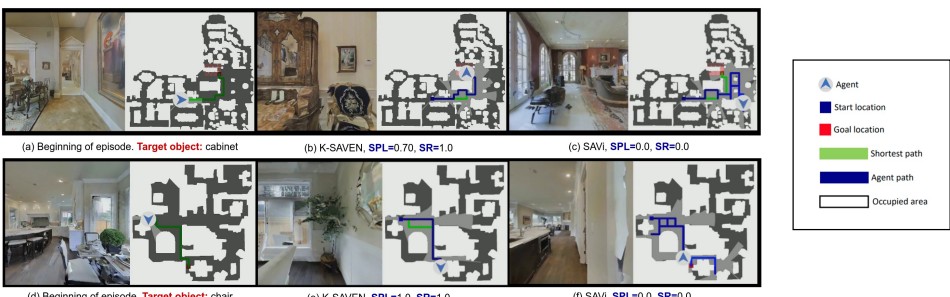

Figure 3: Visualisation of navigation trajectories. Row-wise, we show trajectories and egocentric views for `K-SAVEN` and SAVi on two episodes from the UH/US set. **(a), (d):** beginning of the episode, with the starting pose and view of the agent and the target sounding object. **(b), (e):** `K-SAVEN`'s visual results along with SPL/SR metrics. **(c), (f):** SAVi's visual results along with SPL/SR.

the estimated $\delta_t$ is not reliable enough to provide a consistent weighting scheme across the episode, thus an exponential average with $\delta_t = 0.5$ provides better overall performance.

In heard sounds cases, the agent is familiar with sounds, so vision reasoning is more important. Both *only-GEN$^v$* and *both-GENs* have $GEN^v$; thus, they both perform better than *only-GEN$^b$*, with *only-GEN$^v$* performing slightly better than *both-GENs*, as *only-GEN$^v$* forces the agent to reason only based on vision. For example, in the SH/HS case, the success rate (SR) of *only-GEN$^b$* is 64.4, and the SR of *only-GEN$^v$* and *both-GENs* is 73.2 and 73.0, respectively. In the UH/HS case, the SR of *only-GEN$^b$* is 31.1, and the SR of *only-GEN$^v$* and *both-GENs* is 32.8 and 31.9, respectively. Similarly, in unheard sounds cases, the agent is unfamiliar with sounds, so audio reasoning is more important. Both *only-GEN$^b$* and *both-GENs* have *only-GEN$^b$*; thus they both perform better than *only-GEN$^v$*, with *only-GEN$^b$* performing slightly better than *both-GENs* as *only-GEN$^b$* forces the agent to reason only based on audio. For example, in the SH/US case, the SR of *only-GEN$^v$* is 29.7, and the SR of *only-GEN$^b$* and both-GENs is 31.7 and 30.5, respectively. In the UH/US case, the SR of *only-GEN$^v$* is 21.2, and the SR of *only-GEN$^b$* and *both-GENs* is 23.3 and 22.9, respectively. Furthermore, it is crucial to effectively combine the reasoning capabilities introduced by $GENs$, location prediction, and classification models. Our full model performs better in most ablative cases, indicating that our agent could leverage the reasoning capability using $GENs$ with the memory-based attention mechanism from SMT and generalise to heard and unheard sounds.

**Qualitative results discussion.** We illustrate how our approach qualitatively improves navigation performance, in Fig. 3: we compare `K-SAVEN` and SAVi trajectories on the same episodes, each episode shown row-wise, alongside the episode's corresponding expert trajectory, shown in green. The episodes were obtained from the UH/US set. From these examples, we observe that `K-SAVEN` reaches the goal location in fewer steps, whereas SAVi tends to take more steps and roam throughout the episodes. The latter is supported by the SNA and SPL metrics in Table 1, where `K-SAVEN` achieves higher success in terms of path length (SPL) and number of actions (SNA).

## 7 DISCUSSION AND CONCLUSION

We introduce a framework for leveraging knowledge-enhanced scene priors, in the form of object and region semantics, for the semantic audio-visual navigation task. Notably, we show performance improvements over strong baselines in multiple unseen contexts, particularly in conditions where the agent needed to find novel sounding objects. We also provide a knowledge graph for training models, a curated visual dataset, and a new task definition–all guided towards developing and assessing model generalisation performance in unseen environments. We recognise future improvements of our work, e.g., in the selection of the knowledge resource used for encouraging scene priors in the semantic audio-visual navigation task. We would consider constructing a knowledge resource that characterises sound-object relations (i.e., with descriptions of the sound that is generated by various objects), more befitting of pre-training the acoustic GEN stream. Furthermore, we can consider using scene priors as additive modules on frameworks in other tasks, particularly within the family of embodied multimodal planning. Finally, sounds are not merely a product of individual objects, but of different types of actions and interactions (e.g., sitting, dropping, playing music) that often involve multiple agents and/or objects. Therefore, in future work, we plan to incorporate such semantic knowledge about sounds, objects, actions, and interactions in our knowledge graphs to further improve performance.

## 8 ETHICS STATEMENT

Enabling agents to leverage previous experience and background knowledge, through scene priors, and to better identify goal locations through Direct-to-Reverberant Ratio estimation are of paramount importance in human-machine interaction and robot task-following scenarios. Indeed, systems that are endowed with these capabilities are better-suited for such applications as environmental health monitoring, acoustic anomaly detection, navigation with resource constraints, and others. However, training the scene priors using data that is not general enough for the deployment scenarios could bring bias to the agent's predictions.

## 9 REPRODUCIBILITY STATEMENT

We encourage reproducibility and are committed to enabling broad scientific use of our work, upon acceptance. We will provide code, demonstration videos, our novel knowledge graphs, and our dataset generation scripts—at an anonymised hyperlink, directed to the Reviewers and Areas Chairs, once the submission forums are opened for all submitted papers (as suggested in the ICLR 2023 Author Guide). We provide model implementation details in Section 4, particularly in the paragraph on *Learning and Optimisation*. In Appendix C and Table 5, we further detail the important hyperparameters used in our approach and outline the computing hardware that we used for training and evaluation.

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

## A    ADDITIONAL DETAILS: SIMULATOR, OBJECTS, AND REGIONS

We use SoundSpaces (Chen et al., 2020) to simulate an agent navigating in visually- and acoustically-realistic 3D house environments. The simulator renders sounds at any pair of source (sounding object) and receiver (agent) locations on a uniform grid of nodes spaced by 1 meter. While, SoundSpaces supports two real-world environment scans (Replica (Straub et al., 2019) and Matterport3D (Chang et al., 2017)), we used Matterport3D as it provides a larger number of houses and object-region semantics therein. We use the same 21 object categories as Chen et al. (2021a) for Matterport3D: *chair, table, picture, cabinet, cushion, sofa, bed, chest-of-drawers, plant, sink, toilet, stool, towel, tv monitor, shower, bathtub, counter, fireplace, gym equipment, seating,* and *clothes*. These object categories are visually present in the 24 regions (*balcony, bathroom, bedroom, closet, dining room, entryway/foyer/lobby, familyroom/lounge, hallway, junk, kitchen, laundryroom/mudroom, living room, lounge, meetingroom/conferenceroom, office, other room, porch/terrace/deck, rec/game, spa/sauna, toilet, utilityroom/toolroom,* and *workout/gym/exercise*) of the 85 Matterport3D houses. We use the publicly available sound clips from the experiment performed by Chen et al. (2021a), in which audio clips from `freesound.org` database were used. We generate sound by rendering the specific sound that semantically matches the object at the locations in Matterport3D houses. For example, the water-dropping sound will be associated with the sink in the kitchen.

## B    ADDITIONAL DETAILS: KNOWLEDGE GRAPH

**Knowledge graph construction.** Our knowledge graph captures object-to-object, object-to-region, and region-to-region relations. To compute these relations, we use the semantic labels of objects and regions in Matterport3D. The heuristic we use to find these relations is frequency-based: the main idea is to connect an object with another object if they frequently exist across various regions. Similarly, we connect a region with another region if they both have similar objects placed in them. The resultant knowledge graphs are provided in Tables 3 and 4, which can be represented as adjacency matrices, with an indicator of 1 to characterise a co-occurrence edge between objects, other objects, regions, and other regions. For example, 2 objects (*chair* and *chest-of-drawers*) are connected because they are frequently found in the *bedroom* region. These 2 objects are also frequently found in other regions such as *living room* and *office*. Thus, we can make region-to-region connections by connecting the *bedroom* to the *living room* and the *office* because these regions also frequently contain the same connected objects (*chair* and *chest-of-drawers*) as the *bedroom* region.

**Knowledge graph representation.** Figure 4a illustrates the GloVe embedding space and figure 4b represents the object-region adjacency matrix, both as two-dimensional projections. We reduced the dimension of the GloVe embeddings, for each object and region, into 2 by using ISOMAP Tenenbaum et al. (2000) (shown in Figure 4a). We also reduced the dimension of the vector in the adjacency matrix that encodes the relationship of each object and region with other objects and regions (shown in Figure 4b). As shown in Figure 4, regions and objects are clustered together, and objects found together in houses, such as tables and chairs, are close together.

Alternatively, these graphs can be represented in the same format as existing large-scale commonsense knowledge resources, such as ConceptNet Speer et al. (2017): i.e., as a collection of head **h** / relation **r** / tail **t** triples of the form (**h**, **r**, **t**), with the ConceptNet `LocatedNear` relation for each (h, t)=(object, object) instance pair, the `AtLocation` relation for each (h, t)=(object, region) instance pair, and with the `LocatedNear` relation for each (h, t)=(region, region) instance pair—with saliency weights, based on frequency. Some instances can be further expanded with additional relations, such as `UsedFor`, derived from activity annotations in the region labels. The following examples are taken from the first and tenth rows of Table 3:

> (*bathtub*, `LocatedNear`, *towel*)
> (*bathtub*, `LocatedNear`, *sink*)
> (*bathtub*, `AtLocation`, *bathroom*)
> ...
> (*gym_equipment*, `UsedFor`, *workout*)
> (*gym_equipment*, `AtLocation`, *gym*)
> (*gym_equipment*, `UsedFor`, *exercise*)

## C    ADDITIONAL DETAILS: MODEL IMPLEMENTATION

**Hyperparameters.** For all experiments, we implemented models using the PyTorch deep learning library, version 1.11.0. Table 5 shows the output size of different modules in SAVi Chen et al. (2021a) and different configurations used in the ablation studies of K-SAVEN. In Table 5 "$M$ Size" refers to the size of the vision-based knowledge vector stored in memory $M$, and "Belief Size" refers to the size of the audio-based knowledge vector used to attend to the encoded memory $M_e$. We use ReLU as the activation function $\sigma$ in both GENs. In our experiments, we used $D = 300$ and $F = 64$, respectively, as the input and output feature dimensions in the encoder networks.

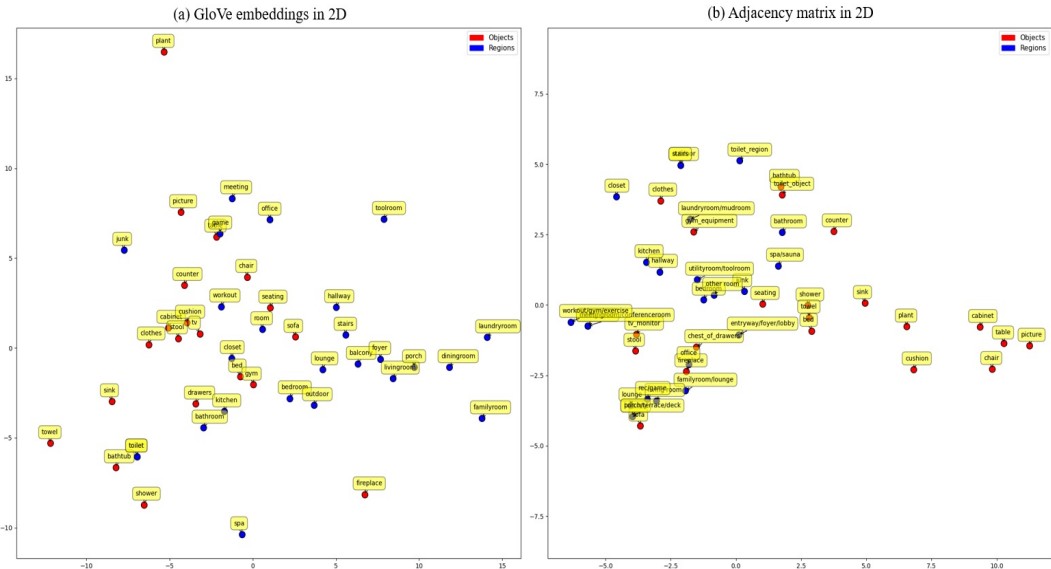

Figure 4: Lower-dimensional projections, illustrating object-region similarity. (a) GloVe embeddings for each object and region into 2D space. (b) Adjacency matrix that encodes the relationship between objects and regions in 2D space. Best viewed in magnification.

Action encoder takes action represented in one-hot vector as input and projects into embedding of size 16. Note that Action encoder is omitted from Fig. 2 (a) for simplicity. The K-SAVEN –only $GEN^b$, K-SAVEN –only $GEN^v$ and K-SAVEN –both $GENs$ configurations are same as the K-SAVEN –full model except the location predictor is not trained to predict direct-to-reverberant ratio (DRR) $\delta_t$. The K-SAVEN –both $GENs + \delta$ is also same as the K-SAVEN –full model except the $\delta_t$ is not fixed to 0.5 and used to estimate the sounding object's location. More specifically, the location is estimate by $l_t = f_\delta(\hat{l}_t, l_{t-1}, \Delta p_t, \delta_t) = \delta_t \hat{l}_t + (1-\delta_t) f_p(l_{t-1}, \Delta p_t)$, where $\delta_t$ is dynamically updated by the location predictor.

**Computing hardware.** For rendering the simulator and performing local agent verification and analysis, we used a single GPU machine, with the following CPU specifications: Intel(R) Core(TM) i5-4690K CPU @ 3.50GHz; 1 CPU, 4 physical cores per CPU, total of 4 logical CPU units. The machine includes a single GeForce GTX TITAN X GPU, with 12.2GB GPU memory. For generating multi-instance experimental results, we used a dual-GPU machine, with the following CPU specifications: Intel(R) Core(TM) i9-9920X CPU @ 3.50GHz; 1 CPU, 12 physical cores per CPU, total 24 logical CPU units. The machine includes two NVIDIA Titan RTX GPUs, each with 24GB GPU memory.

# D ADDITIONAL DETAILS: VISION DATASET

To train the vision classification model $f_c^v$, which given an RGB image predicts a score for objects and regions, we collect a vision dataset using the SoundSpaces simulator as described in Section 4. Initially, we collected 82,828 images across 85 Matterport3D houses, which is the maximum number of images possible as there are a total of 20,707 nodes and 4 rotation angles in SoundSpaces. However, we faced the following challenges with the scans and semantic labelling in the Matterport3D: 1) objects are not clearly visible because of glitches in scans (see RGB image in Figure 5a); 2) object and region semantic labels are improper (see object and region semantic images in Figure 5b and 5c); 3) the way some objects are placed is not common due to the luxurious nature of some houses in Matterport3D (e.g., in scene ID aayBHfsNo7d, there is a big garage, which has a car, a fridge, a table, and chairs; moreover, there is a big pool table in the game room, which is not commonly found in houses), and some objects are not semantically placed (see Figure 6).

To address these challenges, we filtered some images and only used 45,233 images to train our vision classification model $f_c^v$. We use the following filtration criteria: 1) Filter out an image in which 75% of the pixels or more are black (zero value); 2) There are 42 objects in Matterport3D, and we are interested in only 21 objects in our experiments, so we filter out an image if it does not contain any of those 21 objects; 3) Filter out an image if the most frequent object is taking less than 3% of the total pixels in the image; 4) Filter some of the semantic labels of an image based on a threshold (0.18 for object and 0.2 for region). First, for each semantic label in the image, we compute the ratio of its proportion of the pixels to the proportion of the most frequent semantic label in that

Table 3: Relational knowledge graph for spatial object-object interactions

| Sounding objects (21) | Objects (21) | Regions (22) |
|---|---|---|
| bathtub | towel, sink, shower, picture, cabinet, toilet, counter, table, plant | bathroom |
| bed | chair, picture, table, sink, seating, cushion, cabinet, chest_of_drawers, shower, plant, counter, tv_monitor, towel | spa/sauna, junk, bedroom |
| cabinet | clothes, chair, towel, seating, shower, toilet, picture, table, sink, cushion, plant, sofa, counter, bed, chest_of_drawers, bathtub, tv_monitor, stool, fireplace | spa/sauna, bathroom, familyroom/lounge, living room, entryway/foyer/lobby, kitchen, office, utilityroom/toolroom, other room, hallway, laundryroom/mudroom, closet |
| chair | gym_equipment, picture, seating, cushion, table, plant, cabinet, sink, shower, chest_of_drawers, bed, counter, sofa, towel, tv_monitor, stool, fireplace | spa/sauna, familyroom/lounge, living room, junk, entryway/foyer/lobby, kitchen, office, utilityroom/toolroom, bedroom, other room, rec/game, balcony, lounge, porch/terrace/deck, hallway, dining room, meetingroom/conferenceroom, workout/gym/exercise |
| chest_of_drawers | chair, picture, cushion, table, bed, tv_monitor, cabinet | office, bedroom |
| clothes | cabinet, picture | closet |
| counter | towel, cabinet, shower, chair, toilet, picture, sink, cushion, bed, tv_monitor, table, bathtub, plant, stool | bathroom, junk, kitchen, utilityroom/toolroom, laundryroom/mudroom |
| cushion | chair, picture, seating, table, sink, plant, cabinet, shower, chest_of_drawers, bed, sofa, counter, towel, tv_monitor, stool, fireplace | spa/sauna, familyroom/lounge, living room, junk, entryway/foyer/lobby, office, utilityroom/toolroom, bedroom, other room, rec/game, balcony, lounge, porch/terrace/deck |
| fireplace | cushion, table, chair, picture, sofa, plant, stool, cabinet | living room |
| gym_equipment | picture, chair | workout/gym/exercise |
| picture | clothes, gym_equipment, toilet, chair, seating, shower, cushion, towel, cabinet, table, sink, chest_of_drawers, bed, counter, plant, sofa, bathtub, tv_monitor, stool, fireplace | spa/sauna, bathroom, familyroom/lounge, living room, junk, entryway/foyer/lobby, kitchen, office, utilityroom/toolroom, bedroom, other room, rec/game, lounge, hallway, laundryroom/mudroom, closet, dining room, meetingroom/conferenceroom, toilet, workout/gym/exercise |
| plant | chair, picture, sink, towel, table, cushion, shower, toilet, seating, cabinet, sofa, counter, bed, bathtub, tv_monitor, stool, fireplace | spa/sauna, bathroom, familyroom/lounge, living room, junk, entryway/foyer/lobby, rec/game, balcony, porch/terrace/deck |
| seating | chair, table, sink, picture, plant, cabinet, shower, bed, cushion, towel | spa/sauna, entryway/foyer/lobby, other room |
| shower | chair, sink, towel, table, toilet, seating, cabinet, picture, counter, bed, plant, bathtub, cushion | spa/sauna, bathroom |
| sink | cabinet, chair, towel, shower, toilet, seating, picture, table, counter, cushion, bed, tv_monitor, plant, bathtub, stool | spa/sauna, bathroom, junk, kitchen, utilityroom/toolroom, laundryroom/mudroom |
| sofa | chair, picture, cushion, table, plant, cabinet, stool, tv_monitor, fireplace | familyroom/lounge, living room, rec/game, balcony, lounge, porch/terrace/deck |
| stool | cushion, chair, picture, table, cabinet, counter, sofa, plant, sink, tv_monitor, fireplace | familyroom/lounge, living room, kitchen |
| table | chair, towel, picture, seating, shower, toilet, cushion, sink, cabinet, plant, bed, chest_of_drawers, counter, sofa, bathtub, tv_monitor, stool, fireplace | spa/sauna, bathroom, familyroom/lounge, living room, entryway/foyer/lobby, kitchen, office, utilityroom/toolroom, bedroom, other room, rec/game, balcony, lounge, porch/terrace/deck, hallway, dining room, meetingroom/conferenceroom |
| toilet | sink, shower, towel, cabinet, picture, counter, bathtub, table, plant | bathroom, toilet |
| towel | toilet, chair, sink, table, shower, seating, cabinet, picture, counter, bed, plant, bathtub, cushion | spa/sauna, bathroom, toilet |
| tv_monitor | chair, picture, table, cushion, sink, plant, sofa, cabinet, counter, bed, chest_of_drawers, stool | familyroom/lounge, junk, office |

image. Then, semantic labels with ratios less than the threshold are filtered out. 5) For pre-training the vision module, we selected a class cutoff threshold of 0.5, based on a grid search for this hyperparameter, shown in Table 6.

# E    PRE-TRAINED MODELS' PERFORMANCE

**Audio classification model** $f_c^b$: Given an audio signal, our audio classification model, $f_c^b$, predicts the sounding object. We generated 1.5M spectrograms using different source and receiver positions, each corresponding to a sounding object in one of the 85 Matterport3D houses. We used 1,201,147 spectrograms for training and 300,317 spectrograms for testing the audio classification model. We use accuracy $A = \frac{\text{correct predictions}}{\text{total predictions}}\%$ as the metric to evaluate the audio classification performance. We receive $97.3\%$ accuracy when we evaluate the audio classification model on the test set.

**Vision classification model** $f_c^v$: Given an image, our vision classification model $f_c^v$, predicts objects and regions in that image. Out of 45,233 images collected across 85 Matterport3D houses (see Appendix D), we used 36,153 images for training and 9,080 images for testing the vision classification model. We used two metrics to evaluate the performance of the vision classification model. First, we consider exact match ratio, $EMR = \frac{1}{n}\sum_{i=1}^{n}[I(y^{(i)} == \hat{y}^{(i)})]$, where $n$ is number of examples, $y^{(i)}$ and $\hat{y}^{(i)}$ are the true and predicted labels of the $ith$ example, respectively. The EMR calculates the ratio of examples for which the prediction is identical to its ground truth class labels, over all examples. The EMR is always in the range of 0.0-1.0. A high value of the EMR indicates high classification performance. The second metric is the hamming loss $HL = \frac{1}{nL}\sum_{i=1}^{n}\sum_{j=1}^{L}[I(y_j^{(i)} \neq \hat{y}_j^{(i)})]$, where $n$ is number of examples, $L$ is number of classes, and $y_j^{(i)}$

Table 4: Relational knowledge graph for spatial region-region interactions

| Regions (22) | Objects (21) | Other regions (22) |
|---|---|---|
| balcony | chair, plant, cushion, table, sofa | living room, familyroom/lounge, rec/game, porch/terrace/deck |
| bathroom | towel, sink, shower, picture, cabinet, toilet, counter, bathtub, table, plant | spa/sauna |
| bedroom | cushion, picture, chest_of_drawers, bed, chair, table | spa/sauna, office |
| closet | clothes, cabinet, picture | bathroom, hallway, entryway/foyer/lobby, living room, familyroom/lounge, office, kitchen, laundryroom/mudroom, spa/sauna, other room, utilityroom/toolroom |
| dining room | chair, picture, table | bedroom, hallway, entryway/foyer/lobby, living room, familyroom/lounge, office, kitchen, lounge, rec/game, spa/sauna, other room, utilityroom/toolroom, meetingroom/conferenceroom |
| entryway/foyer/lobby | picture, chair, table, plant, cabinet, cushion, seating | spa/sauna |
| familyroom/lounge | cushion, chair, picture, table, plant, sofa, cabinet, tv_monitor, stool | living room |
| hallway | picture, cabinet, chair, table | entryway/foyer/lobby, living room, familyroom/lounge, office, kitchen, spa/sauna, other room, utilityroom/toolroom |
| junk | picture, chair, sink, cushion, counter, plant, bed, tv_monitor | spa/sauna |
| kitchen | cabinet, chair, counter, sink, stool, picture, table | utilityroom/toolroom |
| laundryroom/mudroom | cabinet, counter, picture, sink | bathroom, kitchen, utilityroom/toolroom |
| living room | cushion, table, chair, picture, sofa, plant, stool, fireplace, cabinet | familyroom/lounge |
| lounge | chair, picture, table, cushion, sofa | living room, familyroom/lounge, rec/game |
| meetingroom/conferenceroom | chair, picture, table | bedroom, hallway, dining room, entryway/foyer/lobby, living room, familyroom/lounge, office, kitchen, lounge, rec/game, spa/sauna, other room, utilityroom/toolroom |
| office | chair, table, picture, tv_monitor, chest_of_drawers, cabinet, cushion | familyroom/lounge |
| other room | seating, chair, table, picture, cushion, cabinet | entryway/foyer/lobby, spa/sauna |
| porch/terrace/deck | chair, plant, table, cushion, sofa | balcony, living room, familyroom/lounge, rec/game |
| rec/game | chair, table, cushion, picture, sofa, plant | living room, familyroom/lounge |
| spa/sauna | table, chair, sink, seating, cabinet, shower, picture, bed, plant, towel, cushion | bathroom, entryway/foyer/lobby |
| toilet | toilet, picture, towel | bathroom |
| utilityroom/toolroom | cabinet, chair, picture, table, counter, cushion, sink | kitchen, spa/sauna |
| workout/gym/exercise | gym_equipment, picture, chair | bedroom, hallway, dining room, entryway/foyer/lobby, living room, familyroom/lounge, office, kitchen, lounge, rec/game, spa/sauna, other room, utilityroom/toolroom, junk, meetingroom/conferenceroom |

Table 5: The output size of different modules in SAVi Chen et al. (2021a) and different configurations used in the ablation studies of K-SAVEN.

| Method | Vision Encoder ($f_e^v$) | $GEN^v$ | Audio Encoder ($f_e^b$) | Pose $p_t$ | Action Encoder | M Size | $GEN^b$ | $c_t^b$ | Location $l_t^b$ | Belief Size |
|---|---|---|---|---|---|---|---|---|---|---|
| SAVi Chen et al. (2021a) | 128 | - | 64 | 2 | 16 | 210 | - | 21 | 2 | 23 |
| K-SAVEN –only $GEN^b$ | 128 | - | 64 | 2 | 16 | 210 | 64 | 21 | 2 | 87 |
| K-SAVEN –only $GEN^v$ | 128 | 64 | 64 | 2 | 16 | 274 | - | 21 | 2 | 23 |
| K-SAVEN –both $GENs$ | 128 | 64 | 64 | 2 | 16 | 274 | 64 | 21 | 2 | 87 |
| K-SAVEN –both $GENs + \delta_t$ | 128 | 64 | 64 | 2 | 16 | 274 | 64 | 21 | 2 | 87 |
| K-SAVEN –full model | 128 | 64 | 64 | 2 | 16 | 274 | 64 | 21 | 2 | 87 |

and $\hat{y}_j^{(i)}$ are the true and predicted labels of the $ith$ example and $jth$ class, respectively. The HL measures the average number of false positives and false negatives. For a given class, a low value of the hamming loss indicates that the class is easy to recognize, while a high value shows the opposite. We receive EMR of $0.48$ for objects and $0.68$ for regions when we evaluate the vision classification model on the test set. It is challenging to get a high score on EMR because it does not account for partially correct labels. Comparatively, classifying regions is easier than classifying objects, as indicated by higher EMR. Table 7 shows the hamming loss results. The average HL for all objects is $0.041089785$ and for all regions is $0.017621146$. Lower average HL value for regions compared to that of objects indicate classifying regions is easier than objects, as indicated by EMR results.

# F ADDITIONAL DETAILS: EXPERIMENTS

**Episode specification and success criteria.** An episode of semantic audio-visual embodied navigation task is defined by a house, a start location, and rotation angle of the agent, a goal location, a sounding object, and duration of the audio event. In each episode, the start location and rotation of the agent is randomly selected. For selecting the sounding object, an instance of an object category in the house is also chosen randomly. We define a set of viewpoints within 1 meter of the object's boundary from where the object is visible to the agent. When the agent executes the *Stop* action at any of these viewpoints, the episode will be successfully completed. We sample 367,155 episodes for training and 1000 episodes for each of the testing settings. To select the duration of

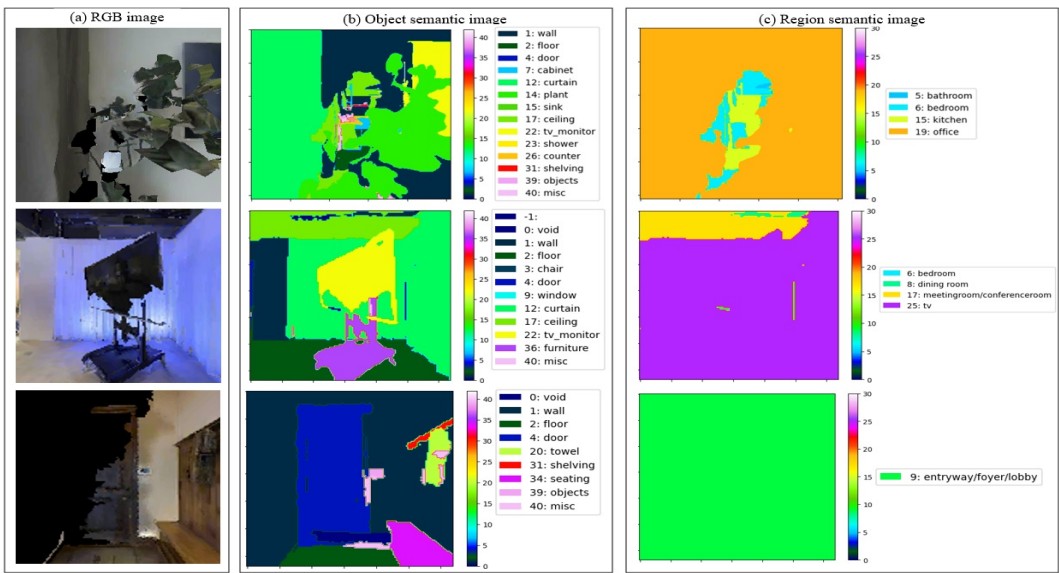

Figure 5: Examples of issues with the scans and semantic labeling in the Matterport3D. The images in the first row correspond to the scene ID `aayBHfsNo7d` and node: (93, 270), images in the second row correspond to the scene ID `2n8kARJN3HM` and node: (8, 180), and images in the third row correspond to the scene ID `1pXnuDYAj8r` and node: (12, 90). (a) shows RGB image examples in which the objects are not clearly visible due to glitches in the scan. (b) shows the semantic labels, and (c) shows the semantic labels; however, these objects and regions are not clearly visible in the corresponding RGB image.

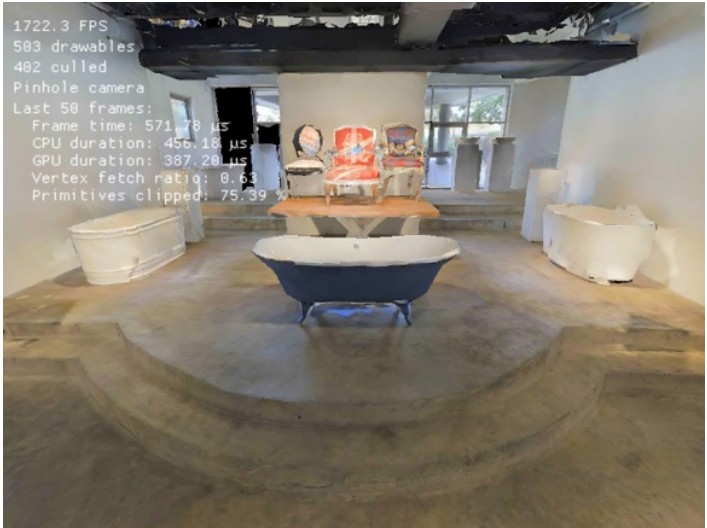

Figure 6: Examples of unusual semantically-placed objects in scene ID `2n8kARJN3HM` of Matterport3D. In the image show, a bathtub is placed in the living room, and chairs are kept on the top of a table, which is unusual placement of these objects.

the audio event, first, we sample a value from a normal distribution with a mean of 15 and a standard deviation of 9, and then we clip this value to limit the duration between 5 and 500 seconds.

**Action space and sensors.** There are 4 actions in the agent's action space: *MoveForward*, *TurnLeft*, *TurnRight*, and *Stop*. *MoveForward* changes the agent's current location to the node in front of it only if that node is reachable without collision. *Stop* can be used by the agent to report sounding objects and terminal the episode. The *TurnLeft*, *TurnRight*, and *Stop* actions can always be executed successfully. There are 4 sensory inputs: egocentric binaural sound (two-channel audio waveforms), RGB image, depth image, and the agent's current pose relative to the starting pose of the episode. The resolutions of the RGB and depth images are $128 \times 128$.

Table 6: Class threshold hyperparameter search, for visual module pre-training. "Shuffle" refers to shuffling the observations, during training time; "EMR" refers to exact match ratio. For "Frozen Pre-trained Params", asterisks (*) refer to the practice of freezing all but the final layer.

| Threshold | Normalised | Image Dimension | Frozen Pre-trained Params | Single / Multiple GPUs | Shuffle | Train Time | EMR: Val | EMR: Test |
|---|---|---|---|---|---|---|---|---|
| 0.5 | Yes | 128x128 | No | Single | No | – | 0.713216 | 1.595627 |
| 0.5 | Yes | 128x128 | No | Multiple | No | 722m 60s | 0.539758 | 1.375566 |
| 0.5 | Yes | 128x128 | No | Single | Yes | 549m 24s | **1.163436** | **1.832025** |
| 0.6 | No | 128x128 | No | Single | No | 951m 60s | 0.663987 | 1.559282 |
| 0.6 | Yes | 128x128 | No | Single | No | 957m 16s | 0.686674 | 1.561979 |
| 0.6 | Yes | 128x128 | Yes* | Single | No | 1257m 6s | 0.154515 | 0.162603 |
| 0.8 | Yes | 128x128 | No | Single | No | 1249m 58s | 0.649119 | 1.548759 |

Table 7: Hamming loss for each object class computed by vision classification model $f_c^v$.

| Sounding objects | Hamming loss | Sounding objects | Hamming loss |
|---|---|---|---|
| bathtub | 0.012775331 | plant | 0.041079298 |
| bed | 0.032709252 | seating | 0.019162996 |
| cabinet | 0.099008814 | shower | 0.010462555 |
| chair | 0.121365644 | sink | 0.010462555 |
| chest_of_drawers | 0.033149779 | sofa | 0.061233483 |
| clothes | 0.003414097 | stool | 0.024339208 |
| counter | 0.035132159 | table | 0.141519830 |
| cushion | 0.048237886 | toilet | 0.003964758 |
| fireplace | 0.034691632 | towel | 0.006497798 |
| gym_equipment | 0.003193833 | tv_monitor | 0.022356829 |
| picture | 0.098127753 | | |

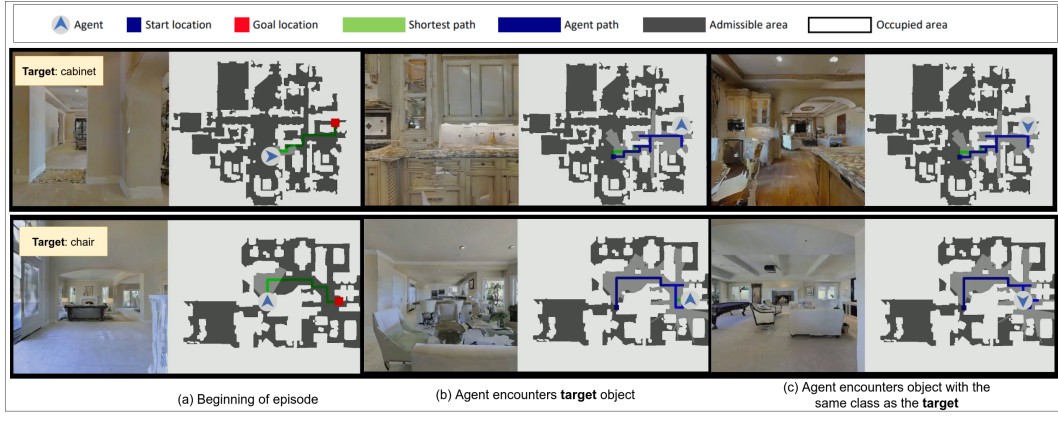

Figure 7: Visualisation of failure case where the agent encounters multiple objects with the same class as the target object. **(a):** beginning of the episode, with the starting pose and view of the agent and the target sounding object. **(b):** K-SAVEN agent encounters the target object, but continues navigating. **(c):** K-SAVEN agent finds a different object with the same class as the target object.

At timestep $t$, the agent must select and execute an action $a_t \in \mathcal{A}$. The goal is to learn a parameterised mapping (e.g., a policy), such that given a sequence of observations $\{O_0, O_1, ..., O_t, ..., O_T\}$, an agent that begins at an initial location in house $h_i \in \mathcal{H}$ can navigate to sounding object $o$.

**Evaluating the memory in SMT.** To evaluate the effectiveness of the memory used in Scene Memory Transformer (SMT), we evaluate our model's performance after the first training stage, in which the memory size $(s_M)$ is one, and the agent uses only the current observations. Table 9 shows the results of K-SAVEN after stage 1 $(s_M = 1)$ and stage 2 $(s_M = 150)$ training. As shown in Table 9, the agent performs consistently better across all metrics in all test cases after stage 2 training, indicating that adding memory helps to navigate efficiently.

Table 8: Hamming loss for each region class computed by vision classification model $f_c^v$.

| Regions | Hamming loss | Regions | Hamming loss |
|---------|--------------|---------|--------------|
| balcony | 0.004955947 | lounge | 0.022026433 |
| bathroom | 0.034471367 | meetingroom/conferenceroom | 0.004074890 |
| bedroom | 0.055506609 | office | 0.017621147 |
| closet | 0.007819383 | other room | 0.008590309 |
| dining room | 0.031497799 | outdoor | 0.002973568 |
| entryway/foyer/lobby | 0.013325991 | porch/terrace/deck | 0.016189428 |
| familyroom/lounge | 0.035022028 | rec/game | 0.011123348 |
| hallway | 0.047136564 | spa/sauna | 0.004405287 |
| junk | 0 | stairs | 0.006277533 |
| kitchen | 0.046035245 | toilet | 0.000440529 |
| laundryroom/mudroom | 0.003634361 | utilityroom/toolroom | 0.002533040 |
| living room | 0.043171808 | workout/gym/exercise | 0.004074890 |

Table 9: Results of SAVi Chen et al. (2021a) and `K-SAVEN` after stage 1 and stage 2 training of SMT, across all dataset splits.

| Method | SEEN HOUSES, HEARD SOUNDS | | | | | SEEN HOUSES, UNHEARD SOUNDS | | | | |
|--------|--------|--------|--------|--------|--------|--------|--------|--------|--------|--------|
| | SR (↑) | SPL (↑) | SNA (↑) | DTG (↓) | SWS (↑) | SR (↑) | SPL (↑) | SNA (↑) | DTG (↓) | SWS (↑) |
| SAVi Chen et al. (2021a) (stage 1) | 55.4 | 41.8 | 44.0 | 3.8 | 24.6 | 15.0 | 9.6 | 9.5 | 10.0 | 7.7 |
| SAVi Chen et al. (2021a) (stage 2) | 67.2 | 53.6 | 52.8 | 1.6 | 37.8 | 21.7 | 15.7 | 13.6 | 6.5 | 12.2 |
| `K-SAVEN` (stage 1) | 38.0 | 27.4 | 22.2 | 4.7 | 23.4 | 14.0 | 9.2 | 8.6 | 11.9 | 5.7 |
| `K-SAVEN` (stage 2) | **70.2** | **52.8** | **53.9** | **1.78** | **31.0** | **37.8** | **27.1** | **25.5** | **5.3** | **17.8** |

| Method | UNSEEN HOUSES, HEARD SOUNDS | | | | | UNSEEN HOUSES, UNHEARD SOUNDS | | | | |
|--------|--------|--------|--------|--------|--------|--------|--------|--------|--------|--------|
| | SR (↑) | SPL (↑) | SNA (↑) | DTG (↓) | SWS (↑) | SR (↑) | SPL (↑) | SNA (↑) | DTG (↓) | SWS (↑) |
| SAVi Chen et al. (2021a) (stage 1) | 20.1 | 14.3 | 13.9 | 13.3 | 7.2 | 11.9 | 7.7 | 6.6 | 11.9 | 5.1 |
| SAVi Chen et al. (2021a) (stage 2) | 32.0 | 21.2 | 18.5 | 10.1 | 18.0 | 15.3 | 10.8 | 8.8 | 10.0 | 8.3 |
| `K-SAVEN` (stage 1) | 16.0 | 11.1 | 10.0 | 12.3 | 7.0 | 12.5 | 8.6 | 7.7 | 11.7 | 4.3 |
| `K-SAVEN` (stage 2) | **35.3** | **24.4** | **22.2** | **8.4** | **18.6** | **34.4** | **23.4** | **21.7** | **6.6** | **14.3** |

# G    ADDITIONAL DETAILS: VISUALISATION OF NAVIGATION RESULTS

**`K-SAVEN`'s Failure Cases.** In Figure 7, we depict one of the common failure scenarios observed within `K-SAVEN`'s navigation results. This is related to cases where multiple objects with the same class as the target robot are in the vicinity of the target sounding object. In many of these episodes, the agent wanders around these objects for an extended period, often encountering the target object, although stopping at the wrong location. Figure 8 shows another failure scenario where the agent gets stuck with surrounding objects in the environment. Typical instances of this failure mode include; the agent getting stuck in narrow halls or colliding with objects within or outside its field-of-view. In these situations, the agent frequently keeps predicting the *Forward* action for a long time before predicting *Stop*. Conversely, it spins around the region where it gets stuck.

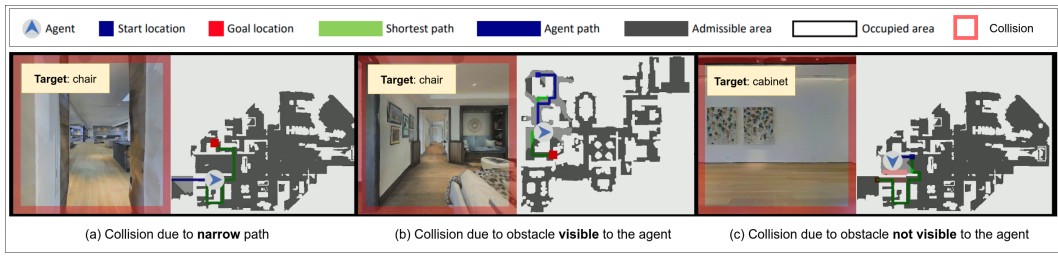

Figure 8: Visualisation of failure case where the agent gets stuck due to collisions with its surroundings. **(a):** collision with a narrow path. **(b):** collision with an object in the field-of-view of the agent. **(c):** collision with an object outside the field-of-view of the agent.

