# OpenReview forum: "Knowledge-driven Scene Priors for Semantic Audio-Visual Embodied Navigation"
_ICLR.cc/2023/Conference — Submitted to ICLR 2023_

### Official Review · Reviewer_8Qq6 · 2022-10-24

**Confidence:** 4
**Correctness:** 2
**Technical Novelty And Significance:** 2
**Empirical Novelty And Significance:** 2
**Recommendation:** 5

**Clarity, Quality, Novelty And Reproducibility:**

Clarity & Quality:
- The paper is a little bit hard to understand and needs more effort in organizing the structure.
- More analysis for ablation studies are necessary.


Novelty:
- The paper does not provide clear and detailed explanations for the novelty of some contributions.
- The new audio-visual navigation task is novel, while the new task does not seem to match with other contributions.


**Strength And Weaknesses:**

Pros: The paper demonstrates the overall architecture clearly in Figure 2.

Cons:
- It is hard for readers to catch up on the key points of the paper. Since almost every subsection only contains one long paragraph, readers might fail to understand the meaning.
- The paper claims to introduce pre-defined semantic information to encode object-object relationships. However, in ICLR 2019, [1] proposed a scene prior graph learned from Visual Genome dataset. It would be helpful if the author could provide a comparison between the proposed semantic information with the semantic knowledge learned in ScenePriors [1].
- The second contribution of the paper is to establish object-object, object-region, and region-region relationships via a knowledge graph. However, in ICLR 2021, [2] proposed a transformer-based architecture to encode similar relationships in visual navigation. The author first misses this reference and second lacks the difference with VTNet [2] in building the aforementioned relationships.
- The paper does not provide convincing analysis for some ablation experiments. For example, in both SH/HS and UH/HS, introducing $GEN^b$ alongside $GEN^v$ will lead to a performance drop compared with only adopting $GEN^v$. Could the author provide some extra explanation for these results?
- Furthermore, it is hard for me to understand why the performance of GEN^s is lower than that of $GEN^b$ in both SH/US and UH/US settings. Could the author please provide more experiments for readers to understand these experiment results?



[1] Yang, W., Wang, X., Farhadi, A., Gupta, A., & Mottaghi, R. (2018). Visual semantic navigation using scene priors. arXiv preprint arXiv:1810.06543.

[2] Du, H., Yu, X., & Zheng, L. (2020, September). VTNet: Visual Transformer Network for Object Goal Navigation. In International Conference on Learning Representations.



**Summary Of The Paper:**

The paper aims to propose two architectural contributions and a new task for semantic audio-visual embodied navigation.

The paper first employs a knowledge graph for encoding object-object, object-region, and region-region relations. Then, the paper introduces multiple auxiliary models to facilitate audio-visual embodied navigation.
Furthermore, the paper proposes a new task of semantic audio-visual embodied navigation. In the new task, the agent is designed to steer toward an object without unheard sound.

The paper also demonstrates performance improvements in all permutation of settings, ie., seen/unseen houses with heard/unheard sounds.


**Summary Of The Review:**

Overall, the paper raises a novel audio-visual navigation task with unheard sounds. However, the other three contributions need more explanations to justify their novelty.

I lean toward rejecting the current version. However, if the author persuades me that all these three contributions are meaningful, I would like to modify my recommendation to be above the acceptance threshold.

Besides the review for this paper, I believe that this kind of open-set setting is interesting and will definitely benefit our community. However, the goal of raising a new task is not because it is a new task. **In my opinion, the goal should be to provide more insights or chances for the community members.** To achieve that, I recommend the author to think questions about some specific problems in open-set tasks and designing methods following corresponding motivations. At final, I hope to see the edited paper in the future.

---

> ### Author Response · Authors · 2022-11-17
> **Response to Reviewer 8Qq6 (1/2)**
>
> Thank you very much for the valuable comments that will significantly improve the quality of our paper. We are happy that the reviewer finds our proposed task novel and the paper’s demonstration of our overall architecture clear. Following are our responses to your concerns and we specifically state what we will update in the paper as a result. Any comments from you on our plan to address your review are welcome.
>
> **Key points.** We list the paper's key contributions in the last paragraph of Section 1 (Introduction). We also briefly explain each module of our framework in the caption of Figure 2. We add several sections in the Appendix that explains several modules in more detail to make them easy to understand for the readers. For example, Appendix A explains the simulator and lists objects and regions used in our experiments. Appendix B demonstrates how object-to-object and region-to-region connections are made with additional, simple examples. Appendix C specifies hyperparameters used for our baseline model SAVi and the several models used in the ablation studies. Appendix D explains the vision dataset in more detail. Appendix E discusses the performance of the classification models. Appendix F discusses some additional experiments we performed. Appendix G discusses the failure cases we observed in our experiments. For your better understanding, we have created a figure for each K-SAVEN configuration used in the ablation studies. We uploaded these figures on Google Drive using an anonymous account. Here are the links for each figure:
> - K-SAVEN–only GEN^b: https://drive.google.com/file/d/10yb3PGmF4QtUq9uMsO7mqlhKMLaDqxmC/
> - K-SAVEN–only GEN^v: https://drive.google.com/file/d/1wiRUy1ihtbspSOOZYM1AjIApxHZqiNid/
> - K-SAVEN–both GENs: https://drive.google.com/file/d/1WroVtTajwMwcnR4SkAzYan_hrtWXdwgK/
> - K-SAVEN–both GENs + δ_t: https://drive.google.com/file/d/1s86mR0PP0kIQReLHNJrTuZW7osT2l1ie/
>
> **Claims + knowledge graph discussion.** Thanks for this note! Our proposed framework leverages object-to-object, object-to-region, as well as region-to-region connections. ScenePriors [1] only provides object-to-object connections; thus, we cannot directly use the semantic knowledge provided by ScenePriors [1] in the audio-visual navigation task. We have refined the text in Section 2 to make this contribution clearer. Moreover, we use a frequency-based heuristic approach, explained in Section 4 (Knowledge graph construction) and Appendix B, based on an automatically computed threshold to find strongly-related objects and regions. However, ScenePriors [1] uses a simpler approach that connects two objects only when their occurrence frequency is more than a specified frequency, which is set to 3. When we tried constructing a knowledge graph using the method in ScenePriors [1], we observed that most of the objects are connected with most of the objects, such that the graph is close to being fully-connected. Thus, we decide to use our frequency-based approach for knowledge graph construction. Before using the knowledge graph in our framework, we performed a preliminary experiment to evaluate our constructed knowledge graph. In this experiment, we plot the vectors in the adjacency matrix that encodes the relationship of each object and region with other objects and regions by reducing the dimension of vectors to 2D. In this plot, we found that regions and objects are clustered together, and objects that are found together are close together (e.g., tables and chairs), as expected. We currently explain this in Appendix B (Knowledge graph representation), but are happy to include more discussion as the reviewer wishes.
>
> **Additional refs.** We thank the reviewer for providing these references. They have some notable differences from our proposed work, however. Du et al. (2021) employ a Visual Transformer Network (VTNet) for the ObjectGoal task. VTNet detects objects in the visual observation, models the relationships among detected object instances, and associates them with image region positions (e.g., bottom and top) to learn directional navigation signals. However, our method uses relationships between objects and regions in the house, encoded as an external knowledge graph, to associate the visual cues with the sound semantics and to navigate effectively in the AudioGoal task. We are happy to add this to the manuscript if the reviewer feels this discussion would be helpful for readers.
>
> (We continue this response in another comment, below)

---

> > ### Author Response · Authors · 2022-11-17
> > **Response to Reviewer 8Qq6 (2/2)**
> >
> > **GEN results.** Thank you for pointing out this observation. We added text discussing this in Section 6 (Ablation studies and analyses). In heard sounds cases, the agent is familiar with sounds, so vision reasoning is more important. Both only-GEN^v and both-GENs have GEN^v; thus they both perform better than only-GEN^b, with only-GEN^v performing slightly better than both-GENs as only-GEN^v forces the agent to reason only based on vision. For example, in the SH/HS case, the success rate (SR) of only-GEN^b is 64.4, and the SR of only-GEN^v and both-GENs is 73.2 and 73.0, respectively. In the UH/HS case, the SR of only-GEN^b is 31.1, and the SR of only-GEN^v and both-GENs is 32.8 and 31.9, respectively. Similarly, in unheard sounds cases, the agent is unfamiliar with sounds, so audio reasoning is more important. Both only-GEN^b and both-GENs have GEN^b; thus they both perform better than only-GEN^v, with only-GEN^b performing slightly better than both-GENs as only-GEN^b forces the agent to reason only based on audio. For example, in the SH/US case, the SR of only-GEN^v is 29.7, and the SR of only-GEN^b and both-GENs is 31.7 and 30.5, respectively. In the UH/US case, the SR of only-GEN^v is 21.2, and the SR of only-GEN^b and both-GENs is 23.3 and 22.9, respectively. Furthermore, it is crucial to effectively combine the reasoning capabilities introduced by GENs, location prediction, and classification models. Our full model performs better in most cases in the ablation study, indicating that our agent could leverage the reasoning capability using GENs with the memory-based attention mechanism using SMT and effectively generalize to heard- and unheard-sounds cases.
> >
> > **Clarity, Quality, Novelty And Reproducibility.** We added several sections, figures, and tables in the appendix to make the paper easier to understand. We also added a paragraph to further discuss the ablation studies.

---

### Official Review · Reviewer_tJCa · 2022-10-24

**Confidence:** 4
**Correctness:** 4
**Technical Novelty And Significance:** 3
**Empirical Novelty And Significance:** 3
**Recommendation:** 6

**Clarity, Quality, Novelty And Reproducibility:**

The paper is well-written, figures and tables are clearly presented. The proposed methods such as knowledge graph, local prediction and direct-to-reverberant ratio estimation are novel in addressing the SAVi problem. The new task supports a more comprehensive and meaningful evaluation of the agent’s performance, which should be beneficial to future research. The authors have stated to release the code and data upon acceptance, which I believe the reproducibility of this work is promising.

**Details Of Ethics Concerns:**

N/A.

**Strength And Weaknesses:**

Strengths:

1.	The paper proposes novel and effective methods and defined a valuable task to address SAVi, which will be highly beneficial to future research in relevant directions. The use of knowledge graph to connect sounds, objects and regions could be an important step towards generalizable audio-visual navigation in novel environments. Results show significant improvement over previous approaches. Moreover, the new task offers a setting for evaluating the agent’s generalization ability on novel objects (and sounds).

2.	It is great to see the comprehensive details presented in the method section and the Appendix, e.g., the knowledge graph construction and representation, including the resultant graphs in Table 3 and Table 4.

3.	Overall, the paper is clearly presented.

Weaknesses:

1.	It is wonderful to see the new experiments added to this paper, but they can go deeper. Since knowledge graph is the key novelty, I expect more experiments on using the knowledge and the graph construction (e.g., the source and type of knowledge, the connection between objects and regions, etc.). Although this work is built based on SAVi and several methods are inherited, it will be valuable to see the influence of the pre-training and classification tasks in the proposed architecture. Moreover, it would greatly strengthen the paper if the method is evaluated on more dataset such as Replica and compare with the baselines.


**Summary Of The Paper:**

The paper studies the Semantic Audio-Visual Navigation (SAVi) and proposes the use of Knowledge-driven scene priors for encoding the object-region relations, spatial knowledge, and background knowledge to address the embodied navigation problem. The method named K-SAVEN incorporates graph encoder networks to model audio and visual inputs, as well as applies scene memory transformer to capture long-term visual-audio dependencies. Numbers of pre-training tasks are defined, and a multimodal dataset is curated to support the learning. Moreover, a new SAVi task is defined to evaluate the agent performance on novel objects and sounds.

**Summary Of The Review:**

This is the second time that I review this paper. Comparing to its previous ICLR and CVPR versions, it is great to see manuscript has been significantly improved and most of my previous concerns are nicely addressed, especially the new experiments in Table 1 (top), Table 2, and Table 7 are added to justify the method and designs, as well as the insightful analysis section.

---

> ### Author Response · Authors · 2022-11-17
> **Response to Reviewer tJCa**
>
> Thank you very much for the valuable comments that will significantly improve the quality of our paper. We are thrilled that the reviewer observes significant improvements in the manuscript compared to previous versions, novelty in the proposed task, and clarity in the writing and in the presentation of the figures and tables! Following are our responses to your concerns and we specifically state what we will update in the paper as a result. Any comments from you on our plan to address your review are welcome.
>
> **Knowledge graph.** Before using the knowledge graph in our framework, we performed preliminary experiments to evaluate our constructed knowledge graph. In one experiment, we plot the vectors in the adjacency matrix that encodes the relationship of each object and region with other objects and regions by reducing the dimension of vectors to 2D. In this plot, we found that regions and objects are clustered together, and objects found together are close together (e.g., tables and chairs), as expected. We provide these results in Appendix B (Knowledge graph representation). Please let us know if you feel that more of this discussion should be moved to the main content.
>
> **Effects of pre-training and classification tasks.** Thank you for the suggestion. GEN^b and GEN^v use the outputs of the pre-trained classification models. In our ablation study (shown in Table 2), we evaluate the performance of K-SAVEN–only GEN^b and K-SAVEN–only GEN^v, which shows the influence of the pre-trained classification models. We evaluated the classification models and have added a new section in Appendix E (Pre-trained models' performance) for discussing the results.
>
> **Replica.** We did consider evaluating our approach on Replica. However, we could not do so in fairness, because we need a dataset that contains semantic labels of objects and regions, and Replica only has semantic labels of objects. We chose Matterport3D for evaluation because it contains semantic labels of objects and regions of 85 houses, whereas Replica contains only object semantic labels of only 18 houses.

---

> > ### Comment · Reviewer_tJCa · 2022-11-18
> > **About pre-training and classification tasks**
> >
> > Thanks the authors for explaining the pre-training and classification tasks, and adding the new Appendix E, expecially the new Table 7 and Table 8. By reading the section, I have a clear understanding of the multi-label classification tasks and their influence to the learning.
> >
> > K-SAVEN–only GEN^b and K-SAVEN–only GEN^v remove both the pre-training and the corresponding GEN, but I am thinking of keeping GEN but remove pre-training, e.g., directly perform a visual/audio to object/region soft-attention to get the class scores. Although not a must-do experiment, I would like to learn about the authors' opinion.
> >
> > Besides, I agree with Reviewer djj6 that if pre-training happened on all sources of sounds (heard and unheard), then there is no true "unheard sound" in testing. Although providing the knowledge about unheard objects does not directly influence the training of GEN and policy network, there is still leak of information about the identity of unheard objects in testing as the classification scores will shift within a fixed given set. I am wondering if such issue can be resolved by totally remove a few objects in both pre-training and training, and adding a new class "unheard (object)" in the pre-training classification.

---

> > > ### Author Response · Authors · 2022-11-18
> > > **RE: About pre-training and classification tasks**
> > >
> > > We appreciate the quick feedback!
> > >
> > > **Removing pre-training.** Regarding the reviewer's question about removing pre-training but keeping the modules in the framework, we have identified a few conditions:
> > >
> > > *Removing pre-training of the classification models, but keeping the classification and GEN modules:* GENs use classification scores from the classification model to associate its visual context with the object/region semantics provided by the knowledge graph, throughout the course of the episode. Without a strong prior, the GEN will attempt spurious associations, which, we hypothesise, will yield worse performance than the baseline.
> > >
> > > *Removing the classification models, but keeping the GENs:* In our framework, using the GENs without the pre-trained classification models would not be possible, since GENs require classification scores that are further associated with the object/region relational semantics provided by the knowledge graph.
> > >
> > > *Removing the knowledge graph, but keeping the GEN modules*: This case will yield similar results as the first case. We hypothesise that, if the object/region semantics were removed, the GENs will inject noise in the downstream parts of the framework (SMT, in the case of GEN^v; policy network, in the case of GEN^b). We actually regard this outcome as being similar to the notion of using a fully-connected knowledge graph — where the knowledge graph would anyway lack the semantically *meaningful* information to inform task-execution.
> > >
> > > **Unheard sounds.** We have not experimented with replacing a few objects with "unheard (object)" in the pre-training classification, mainly because we focus on developing a model that can use knowledge-driven scene priors efficiently. In the original SAVi formulation (Chen et al., 2021), there was semantic leakage in that their method was trained on clips of *all* the objects, then tested on held-out clips from a subset of those *same* objects. We crucially remind the reviewers, here, that it was for this reason that we proposed our novel task and faithfully re-trained the baselines, ablations, and our approach according to this important setting, for fair comparison. We did perform experiments, by comparing the results of a SAVi baseline, that was trained according to their original problem definition (with class leakage) in Chen et al., (2021), against our approach. Even though our approach was only exposed to a strict subset of the objects (without class leakage; according to our novel problem definition), our agent was still able to outperform baselines: our results showed that our agent could reason about unseen houses and unheard sounds and navigate better than the other methods, as it is equipped with the tools to use the prior knowledge to reason about unheard/unseen contexts. Regarding information leakage from the prior, we feel that identities alone are insufficient for achieving performance improvements on the downstream task. Instead, we attribute our model’s performance improvements to its ability to reason more efficiently—using the GENs to *contextualise* the object/region relational semantics (knowledge graph) and the classification scores to the audio-visual navigation task.

---

### Official Review · Reviewer_djj6 · 2022-10-25

**Confidence:** 4
**Correctness:** 3
**Technical Novelty And Significance:** 3
**Empirical Novelty And Significance:** 3
**Recommendation:** 6

**Clarity, Quality, Novelty And Reproducibility:**

I think the paper can be clearer. There are certain mistakes -- for example, in figure 2 (b), they have GCN, but figure 2(a) have GENS. The approach was also quite hard to follow. More importantly, what do the authors mean by unheard sounds is not clear until the very end.

**Strength And Weaknesses:**

Strengths:
- The task definition is interesting. Generalizing to unheard sounds is an interesting split to look at, and I am glad that this is being looked at.

- The proposed approach is somewhat intuitive and seems to work well, specially on unheard sounds.

Weaknesses:
- It's not clear if the pre-training happened on all sources of sounds (seen and unseen). The formulation seems to suggest that the audio classification model was trained on all sources (seen and unseen). I think this is a bit problematic, since it makes the assumption that at test time, no novel sound source will exist.

- If the pre-training involved both seen and unseen sources, then comparison with SAVI is unfair because SAVI presumably didn't have access to these sounds.

- It'd be nice to show results for classification models apart from navigation results. These will help the readers understand how classification accuracy correlates with navigation accuracy.

- I also think it'd be nice to replace certain parts of the model with "oracle" modules. For instance, if the audio classification model, or the location prediction model was perfect, how will the performance get affected. Doing these sort of analyses helps build a much stronger intuition for the task.

**Summary Of The Paper:**

This paper introduces a new kind of audio-visual navigation task which requires agent to generalize to novel audio sources in 3D indoor environments from the Matterport dataset. The proposed approach leverages multiple sources of information of which objects the source is likely coming from, where they might be found, and where the source might be present in the house to tackle the task. In experiments, they show that the proposed approach significantly outperforms existing methods on unseen houses, unheard sounds.


**Summary Of The Review:**

The task is interesting, and the approach is intuitive. But, I think the paper is also missing certain analyses (classification model performance, ablations by replacing with oracle modules) that will further strengthen the paper.

Update after rebuttal: I thank the authors for providing responses to all my concerns and the experimental results for Oracle modules. I am tending towards acceptance for the paper. I am hesitating to give a strong accept is because I would have liked to see experiments where *not all sounds* are heard during the pre-training stage. Since they are building a new benchmark, it'd be nice to push the limits a little bit more and cover the case where a truly novel sound source is present during test stage. For that reason, I am only updating the score to 6.

---

> ### Author Response · Authors · 2022-11-17
> **Response to Reviewer djj6**
>
> Thank you very much for the valuable comments that will significantly improve the quality of our paper. We are happy that the reviewer finds the task interesting and our approach intuitive and performant. Following are our responses to your concerns and we specifically state what we will update in the paper as a result. Any comments from you on our plan to address your review are welcome.
>
> **Sound sources.** Yes, the audio classification model was trained on all sources (heard and unheard). However, during the agent training phase, we only use heard sounds, in which the agent will not hear any sound from the unheard objects and only hear sounds from the heard objects. Moreover, classification score predictions from the pre-trained vision and audio models, along with the word embeddings of those predictions, allow learning the parameters of the Vision and Audio Graph Encoder Networks (GENs), which we use to extract spatial and semantic aware knowledge vectors. Thus, exposing the agent to a subset of sounds during training is essential for enabling it to learn how to use external knowledge. The pre-trained classification models are external modules in our proposed approach and can be trained in any environment. We used Matterport3D for training the pre-trained models because it provides semantic labels of objects and regions in several houses. Our strong baseline, SAVi, also uses a pre-trained audio classification model, which was also trained using Matterport3D on both heard and unheard sounds. We have mentioned this in Section 5 (Baseline models).
>
> **SAVi comparison.** SAVi also uses a pre-trained audio model, which is trained on seen and unseen sources like our method. To clarify this, we have added text in Section 5 (Baseline models).
>
> **Classification model results.** Thank you for your suggestion. We evaluated the classification models and added a new section in Appendix (E. Pre-trained models' performance) for discussing the results.
>
> **Oracle module experiments.** Thank you for your suggestion. Our intuition is that if we incorporate a perfect audio classification model or location prediction model, we will observe an upper bound to the performance on the task. We performed some experiments using our full model and "oracle" modules. When we replace our location prediction model with the perfect location prediction model, we observe improvement in all the test cases. More specifically, the success rate (SR) improved by 2.2% (72.4 vs. 70.2) on SH/US, by 8% (45.8 vs. 37.8) on SH/US, by 4.73% (40.08 vs. 35.35) on UH/HS and by 8.56% (43.00 vs. 34.44) on UH/US. However, when we replaced our audio classification model with the perfect audio classification model, we actually observed a slight drop in performance in most of the test cases. More specifically, the success rate (SR) dropped by 4.4% (65.8 vs. 70.2) on SH/US, improved by 0.62% (38.42 vs. 37.8) on SH/US, dropped by 2.02% (33.33 vs. 35.35) on UH/HS and dropped by 1.72% (32.72-34.44) on UH/US. Our intuition behind this drop is that our GENs are trained to receive classification scores from the classification models; thus, the performance drops when we use perfect classification scores in the form of one-hot vectors. Please let us know if you would like us to include this finding in the paper.
>
> **Clarity, Quality, Novelty And Reproducibility.** Thank you for pointing out the issue with Figure 2 (b); we have updated the figure to address the issue. To clarify unheard sounds, we explain them in the second paragraph of Section 3 (Problem Definition). To further satisfy the reviewer’s recommendations, we have performed additional clarity improvements, throughout the manuscript, in both the main content and appendices.

---

### Decision · Program_Chairs · 2023-01-20

**Decision:**

Reject

**Justification For Why Not Higher Score:**

The issues raised during the review and the virtual meeting are major issues and outweigh the positive points of the paper.

**Justification For Why Not Lower Score:**

N/A

**Metareview: Summary, Strengths And Weaknesses:**

The paper proposes a new task in audio-visual navigation, where the goal is to navigate to unheard sounds. The developed model is based on a knowledge graph that encodes the relationship between objects and regions.
The main strengths of the paper are introducing a novel task and the extensive set of experiments. There are two major weaknesses: (1) The sounds are not unheard. Those objects are used in the pre-training stage. (2) The methods introduced share a lot of similarities with prior work.


**Summary Of Ac-Reviewer Meeting:**

The paper received borderline ratings. So, the AC and the reviewers had a virtual meeting to discuss the paper. While all three reviewers and the AC appreciated the effort behind the paper, the weaknesses are quite major. The reviewers were concerned about using unheard objects during pre-training. That breaks the purpose of the benchmark since those unheard sounds are supposed to be novel. The other major issue that was discussed was the similarity of the method with previous work (as mentioned in the reviews).

The AC also read the reviews, the rebuttal and the paper carefully. The concerns that the reviewers raised are valid concerns and preclude acceptance at this point.